# Beyond Log-concavity: Provable Guarantees for Sampling Multi-modal Distributions using Simulated Tempering Langevin Monte Carlo

**Rong Ge**
Duke University, Computer Science Department
`rongge@cs.duke.edu`

**Holden Lee**
Princeton University, Mathematics Department
`holdenl@princeton.edu`

**Andrej Risteski**
Massachusetts Institute of Technology, Applied Mathematics and IDSS
`risteski@mit.edu`

## Abstract

A key task in Bayesian machine learning is sampling from distributions that are only specified up to a partition function (i.e., constant of proportionality). One prevalent example of this is sampling posteriors in parametric distributions, such as latent-variable generative models. However sampling (even very approximately) can be #P-hard.

Classical results (going back to [BÉ85]) on sampling focus on log-concave distributions, and show a natural Markov chain called *Langevin diffusion* mixes in polynomial time. However, all log-concave distributions are uni-modal, while in practice it is very common for the distribution of interest to have multiple modes. In this case, Langevin diffusion suffers from torpid mixing.

We address this problem by combining Langevin diffusion with *simulated tempering*. The result is a Markov chain that mixes more rapidly by transitioning between different temperatures of the distribution. We analyze this Markov chain for a mixture of (strongly) log-concave distributions of the same shape. In particular, our technique applies to the canonical multi-modal distribution: a mixture of gaussians (of equal variance). Our algorithm efficiently samples from these distributions given only access to the gradient of the log-pdf. To the best of our knowledge, this is the first result that proves fast mixing for multimodal distributions in this setting.

For the analysis, we introduce novel techniques for proving spectral gaps based on decomposing the action of the generator of the diffusion. Previous approaches rely on decomposing the state space as a partition of *sets*, while our approach can be thought of as decomposing the stationary measure as a mixture of *distributions* (a "soft partition").

Additional materials for the paper can be found at `http://tiny.cc/glr17`. Note that the proof and results have been improved and generalized from the precursor at `http://www.arxiv.org/abs/1710.02736`. See Section **??** for a comparison.

# 1  Introduction

Sampling is a fundamental task in Bayesian statistics, and dealing with multimodal distributions is a core challenge. One common technique to sample from a probability distribution is to define a Markov chain with that distribution as its stationary distribution. This general approach is called *Markov chain Monte Carlo*. However, in many practical problems, the Markov chain does not mix rapidly, and we obtain samples from only one part of the support of the distribution.

Practitioners have dealt with this problem through a variety of heuristics. A popular family of approaches involve changing the *temperature* of the distribution. However, there has been little theoretical analysis of such methods. We give provable guarantees for a temperature-based method called *simulated tempering* when it is combined with *Langevin diffusion*.

More precisely, the setup we consider is sampling from a distribution given up to a constant of proportionality. This is inspired from sampling a *posterior* distribution over the latent variables of a latent-variable Bayesian model with known parameters. In such models, the observable variables $x$ follow a distribution $p(x)$ which has a simple and succinct form *given* the values of some latent variables $h$, i.e., the joint $p(h,x)$ factorizes as $p(h)p(x|h)$ where both factors are explicit. Hence, the *posterior* distribution $p(h|x)$ has the form $p(h|x) = \frac{p(h)p(x|h)}{p(x)}$. Although the numerator is easy to evaluate, the denominator $p(x) = \int_h p(h)p(x|h)$ can be NP-hard to approximate even for simple models like topic models [SR11]. Thus the problem is intractable without structural assumptions.

Previous theoretical results on sampling have focused on *log-concave* distributions, i.e., distributions of the form $p(x) \propto e^{-f(x)}$ for a convex function $f(x)$. This is analogous to *convex optimization* where the objective function $f(x)$ is convex. Recently, there has been renewed interest in analyzing a popular Markov Chain for sampling from such distributions, when given gradient access to $f$—a natural setup for the posterior sampling task described above. In particular, a Markov chain called *Langevin Monte Carlo* (see Section 2.1), popular with Bayesian practitioners, has been proven to work, with various rates depending on the precise properties of $f$ [Dal16, DM16, Dal17].

Yet, just as many interesting optimization problems are nonconvex, many interesting sampling problems are not log-concave. A log-concave distribution is necessarily uni-modal: its density function has only one local maximum, which is necessarily a global maximum. This fails to capture many interesting scenarios. Many simple posterior distributions are neither log-concave nor uni-modal, for instance, the posterior distribution of the means for a mixture of gaussians, given a sample of points from the mixture of gaussians. In a more practical direction, complicated posterior distributions associated with deep generative models [RMW14] and variational auto-encoders [KW13] are believed to be multimodal as well.

In this work we initiate an exploration of provable methods for sampling "beyond log-concavity," in parallel to optimization "beyond convexity". As worst-case results are prohibited by hardness results, we must make assumptions on the distributions of interest. As a first step, we consider a mixture of strongly log-concave distributions of the same shape. This class of distributions captures the prototypical multimodal distribution, a mixture of Gaussians with the same covariance matrix. Our result is also robust in the sense that even if the actual distribution has density that is only close to a mixture that we can handle, our algorithm can still sample from the distribution in polynomial time. Note that the requirement that all Gaussians have the same covariance matrix is in some sense necessary: in Appendix K we show that even if the covariance of two components differ by a constant factor, no algorithm (with query access to $f$ and $\nabla f$) can achieve the same robustness guarantee in polynomial time.

## 1.1  Problem statement

We formalize the problem of interest as follows.

**Problem 1.1.** *Let $f : \mathbb{R}^d \to \mathbb{R}$ be a function. Given query access to $\nabla f(x)$ and $f(x)$ at any point $x \in \mathbb{R}^d$, sample from the probability distribution with density function $p(x) \propto e^{-f(x)}$.*

*In particular, consider the case where $e^{-f(x)}$ is the density function of a mixture of strongly log-concave distributions that are translates of each other. That is, there is a base function $f_0 : \mathbb{R}^d \to \mathbb{R}$,*

*centers $\mu_1, \mu_2, \ldots, \mu_m \in \mathbb{R}^d$, and weights $w_1, w_2, \ldots, w_m$ ($\sum_{i=1}^m w_i = 1$) such that*

$$f(x) = -\log\left(\sum_{i=1}^m w_i e^{-f_0(x-\mu_i)}\right), \tag{1}$$

*For notational convenience, we will define $f_i(x) = f_0(x - \mu_i)$.*

The function $f_0$ specifies a basic "shape" around the modes, and the means $\mu_i$ indicate the locations of the modes.

Without loss of generality we assume the mode of the distribution $e^{-f_0(x)}$ is at 0 ($\nabla f_0(0) = 0$). We also assume $f_0$ is twice differentiable, and for any $x$ the Hessian is sandwiched between $\kappa I \preceq \nabla^2 f_0(x)) \preceq KI$. Such functions are called $\kappa$-strongly-convex, $K$-smooth functions. The corresponding distribution $e^{-f_0(x)}$ are strongly log-concave distributions. [1]

## 1.2 Our results

We show that there is an efficient algorithm that can sample from this distribution given just access to $f(x)$ and $\nabla f(x)$.

**Theorem 1.2** (main). *Given $f(x)$ as defined in Equation (1), where the base function $f_0$ satisfies for any $x$, $\kappa I \preceq \nabla^2 f_0(x) \preceq KI$, and $\|\mu_i\| \leq D$ for all $i \in [m]$, there is an algorithm (given as Algorithm 2 with appropriate setting of parameters) with running time* $\text{poly}\left(w_{\min}, D, d, \frac{1}{\varepsilon}, \frac{1}{\kappa}, K\right)$, *which given query access to $\nabla f$ and $f$, outputs a sample from a distribution within TV-distance $\varepsilon$ of $p(x) \propto e^{-f(x)}$.*

Note that importantly the algorithm does *not* have direct access to the mixture parameters $\mu_i, w_i, i \in [n]$ (otherwise the problem would be trivial). Sampling from this mixture is thus non-trivial: algorithms that are based on making local steps (such as the ball-walk [LS93, Vem05] and Langevin Monte Carlo) cannot move between different components of the gaussian mixture when the gaussians are well-separated. In the algorithm we use simulated tempering (see Section 2.2), which is a technique that adjusts the "temperature" of the distribution in order to move between different components.

Of course, requiring the distribution to be *exactly* a mixture of log-concave distributions is a very strong assumption. Our results can be generalized to all functions that are "close" to a mixture of log-concave distributions.

More precisely, assume the function $f$ satisfies the following properties:

$$\exists \tilde{f} : \mathbb{R}^d \to \mathbb{R} \text{ where } \left\|\tilde{f} - f\right\|_\infty \leq \Delta \,, \left\|\nabla \tilde{f} - \nabla f\right\|_\infty \leq \tau \text{ and } \|\nabla^2 \tilde{f}(x) - \nabla^2 f(x)\|_2 \leq \tau, \forall x \in \mathbb{R}^d \tag{2}$$

$$\text{and } \tilde{f}(x) = -\log\left(\sum_{i=1}^m w_i e^{-f_0(x-\mu_i)}\right) \tag{3}$$

$$\text{where } \nabla f_0(0) = 0, \text{ and } \forall x, \kappa I \preceq \nabla^2 f_0(x) \preceq KI. \tag{4}$$

That is, $f$ is within a $e^\Delta$ multiplicative factor of an (unknown) mixture of log-concave distributions. Our theorem can be generalized to this case.

**Theorem 1.3** (general case). *For function $f(x)$ that satisfies Equations (2),(3) and (4), there is an algorithm (given as Algorithm 2 with appropriate setting of parameters) that runs in time* $\text{poly}\left(w_{\min}, D, d, \frac{1}{\varepsilon}, e^\Delta, \tau, \frac{1}{\kappa}, K\right)$, *which given query access to $\nabla f$ and $f$, outputs a sample $x$ from a distribution that has TV-distance at most $\varepsilon$ from $p(x) \propto e^{-f(x)}$.*

Both main theorems may seem simple. In particular, one might conjecture that it is easy to use local search algorithms to find all the modes. However in Section J, we give a few examples to show that such simple heuristics do not work (e.g. random initialization is not enough to find all the modes).

The assumption that all the mixture components share the same $f_0$ (hence when applied to Gaussians, all Gaussians have same covariance) is also necessary. In Section K, we give an example where for a mixture of two gaussians, even if the covariance only differs by a constant factor, any algorithm that achieves similar gaurantees as Theorem 1.3 *must* take exponential time.

## 2   Overview of algorithm

Our algorithm combines *Langevin diffusion*, a chain for sampling from distributions in the form $p(x) \propto e^{-f(x)}$ given only gradient access to $f$ and *simulated tempering*, a heuristic used for tackling multimodality. We briefly define both of these and recall what is known for both of these techniques. For technical prerequisites on Markov chains, the reader can refer to Appendix B.

The basic idea to keep in mind is the following: A Markov chain with local moves such as Langevin diffusion gets stuck in a local mode. Creating a "meta-Markov chain" which changes the temperature (the simulated tempering chain) can exponentially speed up mixing.

### 2.1   Langevin dynamics

Langevin Monte Carlo is an algorithm for sampling from $p \propto e^{-f}$ given access to the gradient of the log-pdf, $\nabla f$.

The continuous version, overdamped Langevin diffusion (often simply called Langevin diffusion), is a stochastic process described by the stochastic differential equation (henceforth SDE)

$$dX_t = -\nabla f(X_t)\, dt + \sqrt{2}\, dW_t \tag{5}$$

where $W_t$ is the Wiener process (Brownian motion). For us, the crucial fact is that Langevin dynamics converges to the stationary distribution given by $p(x) \propto e^{-f(x)}$.

Substituting $\beta f$ for $f$ in (5) gives the Langevin diffusion process for inverse temperature $\beta$, which has stationary distribution $\propto e^{-\beta f(x)}$. Equivalently we can consider the temperature as changing the magnitude of the noise:

$$dX_t = -\nabla f(X_t)dt + \sqrt{2\beta^{-1}}dW_t.$$

Of course algorithmically we cannot run a continuous-time process, so we run a *discretized* version of the above process: namely, we run a Markov chain where the random variable at time $t$ is described as

$$X_{t+1} = X_t - \eta\nabla f(X_t) + \sqrt{2\eta}\xi_k, \quad \xi_k \sim N(0, I) \tag{6}$$

where $\eta$ is the step size. (The reason for the $\sqrt{\eta}$ scaling is that running Brownian motion for $\eta$ of the time scales the variance by $\sqrt{\eta}$.) This is analogous to how gradient descent is a discretization of gradient flow.

#### 2.1.1   Prior work on Langevin dynamics

For Langevin dynamics, convergence to the stationary distribution is a classic result [Bha78]. Fast mixing for log-concave distributions is also a classic result: [BÉ85, BBCG08] show that log-concave distributions satisfy a Poincaré and log-Sobolev inequality, which characterize the rate of convergence—If $f$ is $\alpha$-strongly convex, then the mixing time is on the order of $\frac{1}{\alpha}$. Of course, algorithmically, one can only run a "discretized" version of the Langevin dynamics. Analyses of the discretization are more recent: [Dal16, DM16, Dal17, DK17, DMM18] give running times bounds for sampling from a log-concave distribution over $\mathbb{R}^d$, and [BEL18] give a algorithm to sample from a log-concave distribution restricted to a convex set by incorporating a projection. We note these analysis and ours are for the simplest kind of Langevin dynamics, the overdamped case; better rates are known for underdamped dynamics ([CCBJ17]), if a Metropolis-Hastings rejection step is used ([DCWY18]), and for Hamiltonian Monte Carlo which takes into account momentum ([MS17]).

[RRT17] consider arbitrary non-log-concave distributions with certain regularity and decay properties, but the mixing time is exponential in general; furthermore, it has long been known that transitioning between different modes can take exponentially long, a phenomenon known as meta-stability [BEGK02, BEGK04, BGK05]. The Holley-Stroock Theorem (see e.g. [BGL13]) shows that guarantees for mixing extend to distributions $e^{-f(x)}$ where $f(x)$ is a "nice" function that is close to a convex

function in $L^\infty$ distance; however, this does not address more global deviations from convexity. [MV17] consider a more general model with multiplicative noise.

## 2.2 Simulated tempering

For distributions that are far from being log-concave and have many deep modes, additional techniques are necessary. One proposed heuristic, out of many, is simulated tempering, which swaps between Markov chains that are different temperature variants of the original chain. The intuition is that the Markov chains at higher temperature can move between modes more easily, and hence, the higher-temperature chain acts as a "bridge" to move between modes.

Indeed, Langevin dynamics corresponding to a higher temperature distribution—with $\beta f$ rather than $f$, where $\beta < 1$—mixes faster. (Here, we use terminology from statistical physics, letting $\tau$ denote teh temperature and $\beta = \frac{1}{\tau}$ denote the inverse temperature.) A high temperature flattens out the distribution. However, we can't simply run Langevin at a higher temperature because the stationary distribution is wrong; the simulated tempering chain combines Markov chains at different temperatures in a way that preserves the stationary distribution.

We can define simulated tempering with respect to any sequence of Markov chains $M_i$ on the same space $\Omega$. Think of $M_i$ as the Markov chain corresponding to temperature $i$, with stationary distribution $e^{-\beta_i f}$.

Then we define the simulated tempering Markov chain as follows.

- The *state space* is $\Omega \times [L]$: $L$ copies of the state space (in our case $\mathbb{R}^d$), one copy for each temperature.
- The evolution is defined as follows.
    1. If the current point is $(x, i)$, then evolve according to the $i$th chain $M_i$.
    2. Propose swaps with some rate $\lambda$. When a swap is proposed, attempt to move to a neighboring chain, $i' = i \pm 1$. With probability $\min\{p_{i'}(x)/p_i(x), 1\}$, the transition is successful. Otherwise, stay at the same point. This is a *Metropolis-Hastings step*; its purpose is to preserve the stationary distribution.[2]

The crucial fact to note is that the stationary distribution is a "mixture" of the distributions corresponding to the different temperatures. Namely:

**Proposition 2.1.** *[MP92, Nea96] If the $M_k$, $1 \le k \le L$ are reversible Markov chains with stationary distributions $p_k$, then the simulated tempering chain $M$ is a reversible Markov chain with stationary distribution*

$$p(x, i) = \frac{1}{L} p_i(x).$$

The typical setting of simulated tempering is as follows. The Markov chains come from a smooth family of Markov chains with parameter $\beta \ge 0$, and $M_i$ is the Markov chain with parameter $\beta_i$, where $0 \le \beta_1 \le \ldots \le \beta_L = 1$. We are interested in sampling from the distribution when $\beta$ is large ($\tau$ is small). However, the chain suffers from torpid mixing in this case, because the distribution is more peaked. The simulated tempering chain uses smaller $\beta$ (larger $\tau$) to help with mixing. For us, the stationary distribution at inverse temperature $\beta$ is $\propto e^{-\beta f(x)}$.

### 2.2.1 Prior work on simulated tempering

Provable results of this heuristic are few and far between. [WSH09, Zhe03] lower-bound the spectral gap for generic simulated tempering chains, using a Markov chain decomposition technique due to [MR02]. However, for the Problem 1.1 that we are interested in, the spectral gap bound in [WSH09] is exponentially small as a function of the number of modes. Drawing inspiration from [MR02], we establish a Markov chain decomposition technique that overcomes this.

One issue that comes up in simulated tempering is estimating the partition functions; various methods have been proposed for this [PP07, Lia05].

## 2.3 Main algorithm

Our algorithm is intuitively the following. Take a sequence of inverse temperatures $\beta_i$, starting at a small value and increasing geometrically towards 1. Run simulated tempering Langevin on these temperatures, suitably discretized. Take the samples that are at the $L$th temperature.

Note that there is one complication: the standard simulated tempering chain assumes that we can compute the ratio between temperatures $\frac{p_{i'}(x)}{p_i(x)}$. However, we only know the probability density functions up to a normalizing factor (the partition function). To overcome this, we note that if we use the ratios $\frac{r_{i'}p_{i'}(x)}{r_i p_i(x)}$ instead, for $\sum_{i=1}^{L} r_i = 1$, then the chain converges to the stationary distribution with $p(x, i) = r_i p_i(x)$. Thus, it suffices to estimate each partition function up to a constant factor. We can do this inductively: running the simulated tempering chain on the first $\ell$ levels, we can estimate the partition function $Z_{\ell+1}$; then we can run the simulated tempering chain on the first $\ell + 1$ levels. This is what Algorithm 2 does when it calls Algorithm 1 as subroutine.

A formal description of the algorithm follows.

---

**Algorithm 1** Simulated tempering Langevin Monte Carlo

---

INPUT: Temperatures $\beta_1, \ldots, \beta_\ell$; partition function estimates $\widehat{Z}_1, \ldots, \widehat{Z}_\ell$; step size $\eta$, time $T$, rate $\lambda$, variance of initial distribution $\sigma_0$.
OUTPUT: A random sample $x \in \mathbb{R}^d$ (approximately from the distribution $p_\ell(x) \propto e^{-\beta_\ell f(x)}$).
Let $(i, x) = (1, x_0)$ where $x_0 \sim N(0, \sigma_0^2 I)$.
Let $n = 0, T_0 = 0$.
**while** $T_n < T$ **do**
    Determine the next transition time: Draw $\xi_{n+1}$ from the exponential distribution $p(x) = \lambda e^{-\lambda x}$, $x \geq 0$.
    Let $\xi_{n+1} \leftarrow \min\{T - T_n, \xi_{n+1}\}$, $T_{n+1} = T_n + \xi_{n+1}$.
    Let $\eta' = \xi_{n+1} / \left\lceil \frac{\xi_{n+1}}{\eta} \right\rceil$ (the largest step size $< \eta$ that evenly divides into $\xi_{n+1}$).
    Repeat $\left\lceil \frac{\xi_{n+1}}{\eta} \right\rceil$ times: Update $x$ according to $x \leftarrow x - \eta' \beta_i \nabla f(x) + \sqrt{2\eta'}\xi, \xi \sim N(0, I)$.
    If $T_{n+1} < T$ (i.e., the end time has not been reached), let $i' = i \pm 1$ with probability $\frac{1}{2}$. If $i'$ is out of bounds, do nothing. If $i'$ is in bounds, make a type 2 transition, where the acceptance ratio is $\min \left\{ \frac{e^{-\beta_{i'} f(x)}/\widehat{Z}_{i'}}{e^{-\beta_i f(x)}/\widehat{Z}_i}, 1 \right\}$.
    $n \leftarrow n + 1$.
**end while**
If the final state is $(\ell, x)$ for some $x \in \mathbb{R}^d$, return $x$. Otherwise, re-run the chain.

---

---

**Algorithm 2** Main algorithm

---

INPUT: A function $f : \mathbb{R}^d$, satisfying assumption (2), to which we have gradient access.
OUTPUT: A random sample $x \in \mathbb{R}^d$.
Let $0 \leq \beta_1 < \cdots < \beta_L = 1$ be a sequence of inverse temperatures satisfying (117) and (118).
Let $\widehat{Z}_1 = 1$.
**for** $\ell = 1 \to L$ **do**
    Run the simulated tempering chain in Algorithm 1 with temperatures $\beta_1, \ldots, \beta_\ell$, partition function estimates $\widehat{Z}_1, \ldots, \widehat{Z}_\ell$, step size $\eta$, time $T$, and rate $\lambda$ given by Lemma G.2.
    If $\ell = L$, return the sample.
    If $\ell < L$, repeat to get $n = O(L^2 \ln\left(\frac{1}{\delta}\right))$ samples, and let $\widehat{Z_{\ell+1}} = \widehat{Z}_\ell \left( \frac{1}{n} \sum_{j=1}^{n} e^{(-\beta_{\ell+1} + \beta_\ell) f(x_j)} \right)$.
**end for**

---

# 3 Overview of the proof techniques

We summarize the main ingredients and crucial techniques in the proof, while the full proofs are included in the appendices.

**Step 1:** Define a continuous version of the simulated tempering Markov chain (Definition C.1, Lemma C.2), where transition times are real numbers determined by an exponential weighting time distribution.

**Step 2:** Prove a new *decomposition theorem* (Theorem D.2) for bounding the spectral gap (or equivalently, the mixing time) of the simulated tempering chain we define. This is the main technical ingredient, and also a result of independent interest.

While decomposition theorems have appeared in the Markov Chain literature (e.g. [MR02]), typically one partitions the *state space*, and bounds the spectral gap using (1) the probability flow of the chain inside the individual sets, and (2) between different sets.

In our case, we decompose the *Markov chain* itself; this includes a decomposition of the stationary distribution into components. (More precisely, we show a decomposition theorem on the generator of the tempering chain.) We would like to do this because in our setting, the stationary distribution is exactly a mixture distribution (Problem 1.1).

Our Markov chain decomposition theorem bounds the spectral gap (mixing time) of a simulated tempering chain in terms of the spectral gap (mixing time) of two chains:

1. "component" chains on the mixture components
2. a "projected" chain whose state space is the set of components, and which captures the action of the chain between components as well as the $\chi^2$-divergence between the mixture components.

This means that if the Markov chain on the individual components mixes rapidly, and the "projected" chain mixes rapidly, then the simulated tempering chain mixes rapidly as well. (Note [MR02, Theorem 1.2] does partition into mixture components, but they only consider the special case where they components are laid out in a chain.)

The mixing time of a continuous Markov chain is quantified by a Poincaré inequality.

**Theorem** (Simplified version of Theorem D.2)**.** *Consider the simulated tempering chain $M$ with rate $\lambda = \frac{1}{C}$, where the Markov chain at the $i$th level (temperature) is $M_i = (\Omega, \mathscr{L}_i)$ with stationary distribution $p_i$, for $1 \le i \le L$. Suppose we have a decomposition of the Markov chain at each level, $p_i M_i = \sum_{j=1}^m w_{i,j} p_{i,j} M_{i,j}$, where $\sum_{j=1}^m w_{i,j} = 1$. If each $M_{i,j}$ satisfies a Poincaré inequality with constant $C$, and the $\chi^2$-projected chain $\overline{M}$ satisfies a Poincaré inequality with constant $\overline{C}$, then $M$ satisfies a Poincaré inequality with constant $O(C(1 + \overline{C}))$.*

*Here, the projected chain $\overline{M}$ is the chain on $[L] \times [m]$ with probability flow in the same and adjacent levels given by*

$$\overline{P}((i,j),(i,j')) = \frac{w_{i,j'}}{\chi^2_{\mathrm{sym}}(p_{i,j}, p_{i,j'})} \tag{7}$$

$$\overline{P}((i,j),(i\pm 1,j)) = \frac{\min\left\{\frac{w_{i\pm 1,j}}{w_{i,j}}, 1\right\}}{\chi^2_{\mathrm{sym}}(p_{i,j}, p_{i\pm 1,j'})}, \tag{8}$$

*where $\chi^2_{\mathrm{sym}}(p,q) = \max\{\chi^2(p||q), \chi^2(q||p)\}$.*

The decomposition theorem is the reason why we use a slightly different simulated tempering chain, which is allowed to transition at arbitrary times, with some rate $\lambda$. Such a chain "composes" nicely with the decomposition of the Langevin chain, and allows a better control of the Dirichlet form of the tempering chain, which governs the mixing time.

**Step 3:** Finally, we need to apply the decomposition theorem to our setup, namely a distribution which is a mixture of strongly log-concave distributions. The "components" of the decomposition in

our setup are simply the mixture components $e^{-f_0(x-\mu_j)}$. We rely crucially on the fact that Langevin diffusion on a mixture distribution decomposes into Langevin diffusion on the individual components.

We actually first analyze the *hypothetical* simulated tempering Langevin chain on $\widetilde{p}_i \propto \sum_{j=1}^{m} w_j e^{-\beta_j f_0(x-\mu_j)}$ (Theorem E.1)—i.e., where the stationary distribution for each temperature is a mixture. Then in Lemma E.5 we compare to the *actual* simulated tempering Langevin that we can run, where $p_i \propto p^\beta$. To do this, we use the fact that $p_i$ is off from $\widetilde{p}_i$ by at most $\frac{1}{w_{\min}}$. (This is where the factor of $w_{\min}$ comes in.)

To use our Markov chain decomposition theorem, we need to show two things:

1. The component chains mix rapidly: this follows from the classic fact that Langevin diffusion mixes rapidly for log-concave distributions.

2. The projected chain mixes rapidly: The "projected" chain is defined as having more probability flow between mixture components in the same or adjacent temperatures which are close together in $\chi^2$-divergence.

   By choosing the temperatures close enough, we can ensure that the corresponding mixture components in adjacent temperatures are close in $\chi^2$-divergence. By choosing the highest temperature large enough, we can ensure that all the mixture components at the highest temperature are close in $\chi^2$-divergence.

   From this it follows that we can easily get from any component to any other (by traveling up to the highest temperature and then back down). Thus the projected chain mixes rapidly from the method of canonical paths, Theorem B.4.

Note that the equal variance (for gaussians) or shape (for general log-concave distributions) condition is necessary here. For gaussians with different variance, the Markov chain can fail to mix between components at the highest temperature. This is because scaling the temperature changes the variance of all the components equally, and preserves their ratio (which is not equal to 1).

**Step 4:** We analyze the error from discretization (Lemma F.1), and choose parameters so that it is small. We show that in Algorithm 2 we can inductively estimate the partition functions. When we have all the estimates, we can run the simulated tempering chain on all the temperatures to get the desired sample.

## 4 Conclusion

We initiated a study of sampling "beyond log-convexity." In so doing, we developed a new general technique to analyze simulated tempering, a classical algorithm used in practice to combat multi-modality but that has seen little theoretical analysis. The technique is a new decomposition lemma for Markov chains based on decomposing the *Markov chain* rather than just the *state space*. We have analyzed simulated tempering with Langevin diffusion, but note that it can be applied to any with any other Markov chain with a notion of temperature.

Our result is the first result in its class (sampling multimodal, non-log-concave distributions with gradient oracle access). Admittedly, distributions encountered in practice are rarely mixtures of distributions with the same shape. However, we hope that our techniques may be built on to provide guarantees for more practical probability distributions. An exciting research direction is to provide (average-case) guarantees for probability distributions encountered in practice, such as posteriors for clustering, topic models, and Ising models. For example, the posterior distribution for a mixture of gaussians can have exponentially many terms, but may perhaps be tractable in practice. Another interesting direction is to study other temperature heuristics used in practice, such as particle filters [Sch12, DMHW$^+$12, PJT15, GDM$^+$17], annealed importance sampling [Nea01], and parallel tempering [WSH09].

## Footnotes

[1]On a first read, we recommend concentrating on the case $f_0(x) = \frac{1}{2\sigma^2}\|x\|^2$. This corresponds to the case where all the components are spherical Gaussians with mean $\mu_i$ and covariance matrix $\sigma^2 I$.

[2] This can be defined as either a discrete or continuous Markov chain. For a discrete chain, we propose a swap with probability $\lambda$ and follow the current chain with probability $1 - \lambda$. For a continuous chain, the time between swaps is an exponential distribution with decay $\lambda$ (in other words, the times of the swaps forms a Poisson process). Note that simulated tempering is traditionally defined for discrete Markov chains, but we will use the continuous version. See Definition C.1 for the formal definition.

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
