[Supplementary Material · appendices_nips.pdf]

# Beyond Log-concavity: Provable Guarantees for Sampling Multi-modal Distributions using Simulated Tempering Langevin Monte Carlo—Supplementary materials

**Rong Ge**
Duke University, Computer Science Department
`rongge@cs.duke.edu`

**Holden Lee**
Princeton University, Mathematics Department
`holdenl@princeton.edu`

**Andrej Risteski**
Massachusetts Institute of Technology, Applied Mathematics and IDSS
`risteski@mit.edu`

## Contents

In Section A we restate the main theorems more precisely. In Section B we give some background on Markov chains.

The next five sections are dedicated to the proof of the gaussian case of the main theorem (Theorem A.2). In Section C we formally define the continuous simulated tempering chain. Section D presents our main ingredient: a new Markov chain decomposition theorem. Section E applies it to the gaussian case. Section F bounds the discretization error. Finally, in Section G we combine the previous sections to prove the main theorem, including the analysis of the partition function estimates.

In the next two sections we generalize this result, to the log-concave case (Section H proves Theorem A.3) and the perturbed case (Section I proves Theorem A.4).

In Section J, we give a few examples to show that simple local search heuristics do not work (e.g. random initialization is not enough to find all the modes). In Section K we prove a lower bound when the gaussians have different variances.

Finally, Section L collects a lot of technical calculations and inequalities, particularly inequalities for log-concave distributions that are needed for the proof of the general log-concave case (Section H).

## A  Theorem statements

We restate the main theorems more precisely. First define the assumptions.

**Assumptions A.1.** *The function $f$ satisfies the following. There exists a function $\tilde{f}$ that satisfies the following properties.*

   *1. $\tilde{f}$, $\nabla \tilde{f}$, and $\nabla^2 \tilde{f}$ are close to $f$:*

$$\left\| \tilde{f} - f \right\|_\infty \le \Delta \,, \ \left\| \nabla \tilde{f} - \nabla f \right\|_\infty \le \tau \ \text{and} \ \nabla^2 \tilde{f}(x) \preceq \nabla^2 f(x) + \tau I, \forall x \in \mathbb{R}^d \quad (1)$$

   *2. $\tilde{f}$ is the log-pdf of a mixture:*

$$\tilde{f}(x) = -\log \left( \sum_{i=1}^m w_i e^{-f_0(x-\mu_i)} \right) \quad (2)$$

   *where $\nabla f_0(0) = 0$ and*

      *(a) $f_0$ is $\kappa$-strongly convex: $\nabla^2 f_0(x) \succeq \kappa I$ for $\kappa > 0$.*

      *(b) $f_0$ is $K$-smooth: $\nabla^2 f_0(x) \preceq KI$.*

Our main theorem is the following.

**Theorem A.2** (Main theorem, Gaussian version). *Suppose* $f(x) = -\ln\left(\sum_{j=1}^m w_j \exp\left(-\frac{\|x-\mu_j\|^2}{2\sigma^2}\right)\right)$ *on* $\mathbb{R}^d$ *where* $\sum_{j=1}^m w_j = 1$, $w_{\min} = \min_{1 \le j \le m} w_j > 0$, *and* $D = \max_{1 \le j \le m} \|\mu_j\|$. *Then Algorithm 2 with parameters satisfying* $t, T, \eta^{-1}, \beta_1^{-1}, (\beta_i - \beta_{i-1})^{-1} = \text{poly}\left(\frac{1}{w_{\min}}, D, d, \frac{1}{\sigma^2}, \frac{1}{\varepsilon}\right)$ *produces a sample from a distribution* $p'$ *with* $\|p - p'\|_1 \le \varepsilon$ *in time* $\text{poly}\left(\frac{1}{w_{\min}}, D, d, \frac{1}{\sigma^2}, \frac{1}{\varepsilon}\right)$.

The precise parameter choices are given in Lemma G.2.

Our more general theorem allows the mixture component to come from an arbitrary log-concave distribution $p(x) \propto e^{-f_0(x)}$.

**Theorem A.3** (Main theorem). *Suppose* $p(x) \propto e^{-f(x)}$ *and* $f(x) = -\ln\left(\sum_{i=1}^m w_i e^{-f_0(x-\mu_i)}\right)$ *on* $\mathbb{R}^d$, *where function* $f_0$ *satisfies Assumption A.1(2)* ($f_0$ *is* $\kappa$-*strongly convex,* $K$-*smooth, and has minimum at 0*), $\sum_{i=1}^m w_i = 1$, $w_{\min} = \min_{1 \le i \le m} w_i > 0$, *and* $D = \max_{1 \le i \le n} \|\mu_i\|$. *Then Algorithm 2 with parameters satisfying* $t, T, \eta^{-1}, \beta_1^{-1}, (\beta_i - \beta_{i-1})^{-1} = \text{poly}\left(w_{\min}, D, d, \frac{1}{\varepsilon}, \frac{1}{\kappa}, K\right)$ *produces a sample from a distribution* $p'$ *with* $\|p - p'\|_1 \le \varepsilon$ *in time* $\text{poly}\left(w_{\min}, D, d, \frac{1}{\varepsilon}, \frac{1}{\kappa}, K\right)$.

The precise parameter choices are given in Lemma H.3.

**Theorem A.4** (Main theorem with perturbations). *Keep the setup of Theorem A.3. If instead* $f$ *satisfies Assumption A.1* ($f$ *is* $\Delta$-*close in* $L^\infty$ *norm to the log-pdf of a mixture of log-concave distributions*), *then the result of Theorem A.3 holds with an additional factor of* $\text{poly}(e^\Delta, \tau)$ *in the running time.*

# B  Background on Markov chains

A discrete-time Markov chain on a state space $\Omega$ is defined by transition probabilities $P(x, y)$ where $\sum_{y \in \Omega} P(x, y) = 1$.

A continuous time Markov process is instead defined by $(P_t)_{t \ge 0}$, and a more natural object to consider is the generator.

**Definition B.1.** *A continuous time Markov process is given by* $M = (\Omega, (P_t)_{t \ge 0})$ *where the* $P_t$ *define a random proces* $(X_t)_{t \ge 0}$ *by*

$$\mathbb{P}(X_{s+t} \in A) = P_t(x, A) := \int_A P_t(x, y)\, dy.$$

*Define the action of* $P_t$ *on functions by*

$$(P_t g)(x) = \mathbb{E}_{y \sim P(x, \cdot)} g(y) = \int_\Omega P(x, y)\, dy.s \tag{3}$$

*A* **stationary distribution** *is* $p(x)$ *such that if* $X_0 \sim p$, *then* $X_t \sim p$ *for all* $t$.

*Define the* **generator** $\mathscr{L}$ *by*

$$\mathscr{L}g = \lim_{t \searrow 0} \frac{P_t g - g}{t}. \tag{4}$$

*If* $p$ *is the unique stationary distribution, define the* **Dirichlet form** *and the variance by*

$$\mathscr{E}_M(g, h) = -\langle g, \mathscr{L}h \rangle_p \tag{5}$$

$$\text{Var}_p(g) = \left\| g - \int_\Omega gp\, dx \right\|_p^2 \tag{6}$$

Note that in order for $(P_t)_{t \ge 0}$ to be a valid Markov process, it must be the case that $P_t P_u g = P_{t+u} g$, i.e., the $(P_t)_{t \ge 0}$ forms a **Markov semigroup**.

We will use the shorthand $\mathscr{E}(g) := \mathscr{E}(g, g)$.

**Definition B.2.** *A continuous Markov process satisfies a **Poincaré inequality with constant** $C$ if for all $g$ such that $\mathscr{E}_M(g)$ is defined (finite),*[1]

$$\mathscr{E}_M(g) \geq \frac{1}{C} \operatorname{Var}_p(g). \tag{7}$$

This is another way of saying that the spectral gap of the Markov process satisfies $\operatorname{Gap}(M) \geq \frac{1}{C}$.

For Langevin diffusion with stationary distribution $p$,

$$\mathscr{E}_M(g) = \|\nabla g\|_p^2. \tag{8}$$

Since this depends in a natural way on $p$, we will also write this as $\mathscr{E}_p(g)$. A Poincaré inequality for Langevin diffusion thus takes the form

$$\mathscr{E}_p(g) = \int_\Omega \|\nabla g\|^2 \, p \, dx \geq \frac{1}{C} \operatorname{Var}_p(g). \tag{9}$$

We have the following classical result.

**Theorem B.3** ([BGL13])**.** *Let $f$ be $\rho$-strongly convex and differentiable. Then for $g \in \mathcal{D}(\mathscr{E}_p)$, $p \propto e^{-f}$ satisfies the Poincaré inequality*

$$\mathscr{E}_p(g) \geq \rho \operatorname{Var}_p(g).$$

In particular, this holds for $f(x) = \frac{\|x-\mu\|^2}{2}$ with $\rho = 1$, giving a Poincaré inequality for the gaussian distribution.

A spectral gap, or equivalently a Poincaré inequality, implies rapid mixing:

$$\|P_t g - \mathbb{E}_p g\|_2 \leq e^{-t \operatorname{Gap}(M)} = e^{-\frac{t}{C}}. \tag{10}$$

The following gives one way to prove a Poincaré inequality.

**Theorem B.4** (Comparison theorem using canonical paths, [DSC93])**.** *Let $P : \Omega \times \Omega \to \mathbb{R}$ be a function with $P(x,y) \geq 0$ for $y \neq x$ and $\sum_{y \in \Omega} P(x,y) = 1$. (Think of $L$ as a matrix in $\mathbb{R}^{\Omega \times \Omega}$ that operates on functions $g : \Omega \to \mathbb{R}$, i.e., $g \in \mathbb{R}^\Omega$.) Let $L = I - P$, so that $L(x,y) = P(x,y)$ for $y \neq x$ and $L(x,x) = -\sum_{y \in \Omega} P(x,y)$.*

*Consider the Markov chain generated by $L$ ($L$ acts as $Lg(j) = \sum_{k \neq j}[g(k) - g(j)]P(j,k)$); let its Dirichlet form be $\mathscr{E}(g,g) = -\langle g, Lg \rangle$.*

*Suppose each pair $x, y \in \Omega$, $x \neq y$ is associated with a path $\gamma_{x,y}$. Define the congestion to be*

$$\rho(\gamma) = \max_{z,w \in \Omega, z \neq w} \left[ \frac{\sum_{\gamma_{x,y} \ni (z,w)} |\gamma_{x,y}| p(x)p(y)}{p(z)L(z,w)} \right].$$

*Then*

$$\operatorname{Var}_p(g) \leq \rho(\gamma)\mathscr{E}(g,g).$$

Note this theorem is more commonly stated in terms of discrete-time Markov chains, where we think of $P$ as the transition probabilities. In the continuous case it's more natural to look at $L$.

## C  Simulated tempering

First we define a continuous version of the simulated tempering Markov chain (Definition C.1). Unlike the usual definition of a simulated tempering chain in the literature, the transition times can be arbitrary real numbers. Our definition falls out naturally from writing down the generator $\mathscr{L}$ as a combination of the generators for the individual chains and for the transitions between temperatures (Lemma C.2). Because $\mathscr{L}$ decomposes in this way, the Dirichlet form $\mathscr{E}$ will be easier to control in Theorem D.2

**Definition C.1.** *Let $M_i, i \in [L]$ be a sequence of continuous Markov chains with state space $\Omega$. Let $r_i$, $1 \le i \le L$ satisfy*

$$r_i > 0, \quad \sum_{i=1}^{L} r_i = 1.$$

*Define the **continuous simulated tempering Markov chain** $M_{st}$ with rate $\lambda$ and relative probabilities $r_i$ as follows.*

*The states of $M_{st}$ are $[L] \times \Omega$.*

*For the evolution, let $(T_n)_{n \ge 0}$ be a Poisson point process on $\mathbb{R}_{\ge 0}$ with rate $\lambda$, i.e., $T_0 = 0$ and*

$$\mathbb{P}(T_{n+1} - T_n = t | T_1, \dots, T_n) = \lambda e^{-\lambda t}. \tag{11}$$

*If the state at time $T_n$ is $(i, x)$, then the Markov chain evolves according to $M_i$ on the time interval $[T_n, T_{n+1})$. The state $X_{T_{n+1}}$ at time $T_{n+1}$ is obtained from the state $X_{T_{n+1}}^- := \lim_{t \to T_{n+1}^-} X_t$ by a "Type 2" transition: If $X_{T_{n+1}}^- = (i, x)$, then transition to $(j = i \pm 1, x)$ each with probability*

$$\frac{1}{2} \min \left\{ \frac{r_j p_j(x)}{r_i p_i(x)}, 1 \right\}$$

*and stay at $(i, x)$ otherwise. (If $j$ is out of bounds, then don't move.)*

**Lemma C.2.** *Let $M_i, i \in [L]$ be a sequence of continuous Markov chains with state space $\Omega$, generators $\mathscr{L}_i$, and unique stationary distributions $p_i$. Then the continuous simulated tempering Markov chain $M_{st}$ with rate $\lambda$ and relative probabilities $r_i$ has generator $\mathscr{L}$ defined by the following equation, where $g = (g_1, \dots, g_L) \in L^2([L] \times \Omega)$:*

$$(\mathscr{L}g)(i, y) = (\mathscr{L}_i g_i)(y) + \frac{\lambda}{2} \left( \sum_{\substack{1 \le j \le L \\ j = i \pm 1}} \int_\Omega \min \left\{ \frac{r_j p_j(x)}{r_i p_i(x)}, 1 \right\} (g_j(x) - g_i(x)) \, dx \right). \tag{12}$$

The corresponding Dirichlet form is

$$\mathscr{E}(g, g) = -\langle g, \mathscr{L}g \rangle = \sum_{i=1}^{L} r_i \mathscr{E}_i(g_i, g_i) + \frac{\lambda}{2} \sum_{\substack{1 \le i, j \le L \\ j = i \pm 1}} \int_\Omega r_i p_i(x) \min \left\{ \frac{r_j p_j(x)}{r_i p_i(x)}, 1 \right\} (g_i(x)^2 - g_i(x)g_j(x)) \tag{13}$$

$$= \sum_{i=1}^{L} r_i \mathscr{E}_i(g_i, g_i) + \frac{\lambda}{4} \sum_{\substack{1 \le i, j \le L \\ j = i \pm 1}} \int_\Omega (g_i - g_j)^2 \min\{r_i p_i, r_j p_j\} \, dx \tag{14}$$

where $\mathscr{E}_i(g_i, g_i) = -\langle g_i, \mathscr{L}_i g_i \rangle_{p_i}$.

*Proof.* Continuous simulated tempering is a Markov chain because the Poisson process is memoryless. We show that its generator equals $\mathscr{L}$. Let $F$ be the operator which acts by

$$Fg(x, i) = g_i(x) + \frac{1}{2} \sum_{\substack{1 \le j \le L \\ j = i \pm 1}} \min \left\{ \frac{r_j p_j(x)}{r_i p_i(x)}, 1 \right\} (g_j(x) - g_i(x)) \tag{15}$$

Let $N_t = \max \{n : T_n \le t\}$. Let $P_{j,t}$ be such that $(P_{j,t}g)(x) = \mathbb{E}_{M_j}[g(x_t)|x_0 = x]$, the expected value after running $M_j$ for time $t$, and let $P_t$ the same operator for $M$.

We have (here, $\delta_j(i) = \mathbb{1}_{i=j}$ is a function on $[L]$)

$$P_t g = \mathbb{P}(N_t = 0) \sum_{j=1}^{L} \delta_j \times P_{j,t} g_j + \int_0^t \mathbb{P}(t_1 = s, N_t = 1) P_{t-s} F P_s g \, ds + \mathbb{P}(N_t \ge 2) h. \tag{16}$$

where $\|h\|_1 \leq \|g\|_1$. By basic properties of the Poisson process, $\mathbb{P}(N_t = 0) = 1 - \lambda t + O(t^2)$, $\mathbb{P}(t_1 = s, N_t = 1) = \lambda + O(t)$ for $0 \leq s \leq t$, and $\mathbb{P}(N \geq 2) = O(t^2)$, so

$$\frac{d}{dt}(P_t g)|_{t=0} = -\lambda \underbrace{\sum_{j=1}^{L} \delta_j \times P_{j,t} g_j}_{g} + \sum_{j=1}^{L} \delta_j \times \mathscr{L}_j g_j + \lambda F g = \mathscr{L}g. \tag{17}$$

$\square$

# D Markov chain decomposition theorems

## D.1 $\chi^2$ Gap-Product Theorem

The main Markov chain decomposition theorem we use to prove Theorem A.2 is Theorem D.2. As a warm-up, we first prove a simpler gap-product theorem, Theorem D.1.

Note that Theorem D.1 is incomparable to the original by [MR02]. In their theorem, they decompose $\Omega$ into a partition, the projected chain is on the sets of the partition, and the transition probabilities are the probabilities of transitioning between the two sets. Here, we decompose into a sum of probability distributions instead (a soft partition), and the transition probabilities in the projected chain is determined by the $\chi^2$-distance between the distributions.

We state the theorem for continuous Markov chains; it works for discrete chains with the appropriate modifications.

**Theorem D.1** ($\chi^2$ Gap-Product Theorem). *Let $M = (\Omega, \mathscr{L})$ be a continuous-time Markov chain with stationary distribution $p$ and Dirichlet form $\mathscr{E}(g,g) = -\langle g, \mathscr{L}g \rangle$. Suppose the following hold.*

1. *There is a decomposition*

$$p\mathscr{L} = \sum_{j=1}^{m} w_j p_j \mathscr{L}_j \tag{18}$$

$$p = \sum_{j=1}^{m} w_j p_j. \tag{19}$$

   *where $\mathscr{L}_j$ is the generator for some Markov chain $M_j$ on $\Omega$ with stationary distribution $p_j$.*

2. *(Mixing for each $M_j$) The Dirichlet form $\mathscr{E}_j(f,g) = -\langle f, \mathscr{L}_j g \rangle_{p_j}$ satisfies the Poincaré inequality*

$$\mathrm{Var}_{p_j}(g) \leq C\mathscr{E}_j(g,g). \tag{20}$$

3. *(Mixing for projected chain) Define the $\chi^2$-projected chain $\overline{M}$ as the Markov chain on $[m]$ generated by $\overline{\mathscr{L}}$, where $\overline{\mathscr{L}}$ acts on $\overline{g} \in L^2([m])$ by*

$$\overline{\mathscr{L}}\overline{g}(j) = \sum_{1 \leq k \leq m, k \neq j} [\overline{g}(k) - \overline{g}(j)]\overline{P}(j,k) \tag{21}$$

$$\text{where } \overline{P}(j,k) = \frac{w_k}{\max\{\chi^2(p_j\|p_k), \chi^2(p_k\|p_j)\}}. \tag{22}$$

   *(I.e., the rate of diffusion from $j$ to $k$ is given by $\overline{P}(j,k)$.) Let $\overline{p}$ be the stationary distribution of $\overline{M}$. Then $\overline{M}$ satisfies the Poincaré inequality*

$$\mathrm{Var}_{\overline{p}}(\overline{g}) \leq \overline{C}\overline{\mathscr{E}}(g,g). \tag{23}$$

*Then $M$ satisfies the Poincaré inequality*

$$\mathrm{Var}_p(g) \leq C\left(1 + \frac{\overline{C}}{2}\right)\mathscr{E}(g,g). \tag{24}$$

*Proof.* First, note that the stationary distribution $\overline{p}$ of $\overline{M}$ is given by $\overline{p}(j) = w_j$, because $w_j \overline{P}(j,k) = w_k \overline{P}(k,j)$. (Note that the reaston $\overline{P}$ has a maximum of $\chi^2$ divergences in the denominator is to make this "detailed balance" condition hold.)

Given $g \in L^2(\Omega)$, define $\overline{g} \in L^2([m])$ by $\overline{g}(j) = \mathbb{E}_{p_j} g$. Then

$$\text{Var}_p(g) = \sum_{j=1}^{m} w_j \int (g - \mathbb{E}g)^2 p_j \tag{25}$$

$$= \sum_{j=1}^{m} w_j \int (g - \underset{p_j}{\mathbb{E}}\, g)^2 p_j + \sum_{j=1}^{m} w_j (\underset{p_j}{\mathbb{E}}\, g - \underset{p}{\mathbb{E}}\, g)^2 \tag{26}$$

$$\leq \sum_{j=1}^{m} w_j \int (g - \underset{p_j}{\mathbb{E}}\, g)^2 p_j + \sum_{j=1}^{m} \overline{p}(j)(\overline{g}(j) - \underset{\overline{p}}{\mathbb{E}}\, \overline{g})^2 \tag{27}$$

$$\leq C \sum_{j=1}^{m} w_j \mathscr{E}_{p_j}(g,g) + \text{Var}_{\overline{p}}(\overline{g}) \tag{28}$$

$$\leq C \mathscr{E}(g,g) + \overline{C}\overline{\mathscr{E}}(\overline{g},\overline{g}). \tag{29}$$

Now

$$\overline{\mathscr{E}}(\overline{g},\overline{g}) = \frac{1}{2} \sum_{j=1}^{m}\sum_{k=1}^{m} (\overline{g}(j) - \overline{g}(k))^2 w_j \overline{P}(j,k) \tag{30}$$

$$\leq \frac{1}{2} \sum_{j=1}^{m}\sum_{k=1}^{m} (\overline{g}(j) - \overline{g}(k))^2 w_j \frac{w_k}{\chi^2(p_j||p_k)} \tag{31}$$

$$\leq \frac{1}{2} \sum_{j=1}^{m}\sum_{k=1}^{m} \text{Var}_{p_j}(g) w_j w_k \qquad \text{by Lemma L.1} \tag{32}$$

$$\leq \frac{1}{2} \sum_{j=1}^{m} w_j C \mathscr{E}_j(g,g) = \frac{C}{2} \mathscr{E}(g,g). \tag{33}$$

Thus

$$(29) \leq C \mathscr{E}(g,g) + \frac{\overline{C}C}{2} \mathscr{E}(g,g) \tag{34}$$

as needed. □

## D.2 $\chi^2$ Gap-Product Theorem for Simulated Tempering

**Theorem D.2** ($\chi^2$ Gap-Product Theorem for Simulated Tempering). *Consider simulated tempering $M$ with Markov chains $M_i = (\Omega, \mathscr{L}_i)$, $1 \leq i \leq L$. Let the stationary distribution of $M_i$ be $p_i$, the relative probabilities be $r_i$, and the rate be $\lambda$. Let the Dirichlet form be $\mathscr{E}_i(g,h) = -\langle g, \mathscr{L}_i h\rangle$.*

*Represent a function $g \in [L] \times \Omega$ as $(g_1, \ldots, g_L)$. Let $p$ be the stationary distribution on $[L] \times \Omega$, $\mathscr{L}$ be the generator, and $\mathscr{E}(g,h) = -\langle g, \mathscr{L}h\rangle$ be the Dirichlet form.*

*Suppose the following hold.*

1. *There is a decomposition*

$$p_i \mathscr{L}_i = \sum_{j=1}^{m_i} w_{i,j} p_{i,j} \mathscr{L}_{ij} \tag{35}$$

$$p_i = \sum_{j=1}^{m_i} w_{i,j} p_{i,j}. \tag{36}$$

*where $\mathscr{L}_{ij}$ is the generator for some Markov chain $M_{i,j}$ on $\{i\} \times \Omega$ with stationary distribution $p_{i,j}$.*

2. *(Mixing for each $M_{i,j}$)* $M_{i,j}$ *satisfies the Poincaré inequality*

$$\text{Var}_{p_{i,j}}(g) \le C \mathscr{E}_{i,j}(g,g). \tag{37}$$

3. *(Mixing for projected chain) Define the $\chi^2$-projected chain $\overline{M}$ as the Markov chain on $[n]$ generated by $\overline{\mathscr{L}}$, where $\overline{\mathscr{L}}$ acts on $\overline{g} \in L^2([n])$ by*

$$\overline{\mathscr{L}}\overline{g}(i,j) = \sum_{i=1}^{L}\sum_{j=1}^{m_i}\sum_{i'=1}^{L}\sum_{j'=1}^{m_{i'}}[\overline{g}(i',j') - \overline{g}(i,j)]\overline{P}((i,j),(i',j')) \tag{38}$$

*where* $\overline{P}((i,j),(i',j')) = \begin{cases} \frac{w_{i,j'}}{\max\{\chi^2(p_{i,j}\|p_{i,j'}),\chi^2(p_{i,j'}\|p_{i,j})\}}, & i = i' \\ \frac{\min\left\{\frac{r_{i'}w_{i',j'}}{r_i w_{i,j}},1\right\}}{\max\{\chi^2(p_{i,j}\|p_{i',j'}),\chi^2(p_{i',j'}\|p_{i,j})\}}, & i' = i \pm 1 \\ 0, & \text{else.} \end{cases} \tag{39}$

*Let $\overline{p}$ be the stationary distribution of $\overline{M}$. Then $\overline{M}$ satisfies the Poincaré inequality*

$$\text{Var}_{\overline{p}}(\overline{g}) \le \overline{C}\overline{\mathscr{E}}(g,g). \tag{40}$$

*Then $M$ satisfies the Poincaré inequality*

$$\text{Var}(g) \le \max\left\{C\left(1 + \frac{13}{2}\overline{C}\right), \frac{6\overline{C}}{\lambda}\right\}\mathscr{E}(g,g). \tag{41}$$

The proof starts as before. The new part is that the projected Markov chain involves transitions between levels, so $\overline{\mathscr{E}}$ will involve taking differences of functions on adjacent levels. That part will be bounded by the second term in (13).

*Proof.* First, note that the stationary distribution $\overline{p}$ of $\overline{M}$ is given by $\overline{p}((i,j)) = r_i w_{i,j}$, because $\overline{P}((i,j),(i',j'))r_i w_{i,j} = \overline{P}((i',j'),(i,j))r_{i'}w_{i',j'}$.

Given $g \in L^2([L] \times \Omega)$, define $\overline{g} \in L^2\left(\bigcup_{i=1}^{L}(\{i\} \times [m_i])\right)$ by $\overline{g}(i,j) = \mathbb{E}_{p_{i,j}} g_i$.

$$\text{Var}_p(g) = \sum_{i=1}^{L}\sum_{j=1}^{m_i} r_i w_{i,j} \int (g - \mathbb{E}_p g)^2 p_j \tag{42}$$

$$= \sum_{i=1}^{L}\sum_{j=1}^{m_i} r_i w_{i,j}\left[\left(\int (g_i - \mathbb{E}_{p_{i,j}} g_i)^2 p_{i,j}\right) + (\mathbb{E}_{p_{i,j}} g_i - \mathbb{E}_p g)^2\right] \tag{43}$$

$$\le C\sum_{i=1}^{L}\sum_{j=1}^{m_i} r_i w_{i,j}\mathscr{E}_{i,j}(g_i,g_i) + \text{Var}_{\overline{p}}(\overline{g}) \tag{44}$$

$$\le C\sum_{i=1}^{L} r_i \mathscr{E}_i(g_i,g_i) + \overline{C}\overline{\mathscr{E}}(\overline{g},\overline{g}). \tag{45}$$

Now $\overline{\mathscr{E}}$ has two terms; the first is bounded in the same way as the Theorem D.1.

$$\overline{\mathscr{E}}(\overline{g},\overline{g}) = \frac{1}{2}\underbrace{\sum_{i=1}^{L}\sum_{j=1}^{n}\sum_{j'=1}^{n}(\overline{g}(i,j) - \overline{g}(i,j'))^2 r_i w_{i,j}\overline{P}((i,j),(i,j'))}_{A} \tag{46}$$

$$+ \frac{1}{2}\underbrace{\sum_{i=1}^{L}\sum_{j=1}^{m_i}\sum_{\substack{i'=i\pm 1 \\ 1 \le i' \le L}}\sum_{j=1}^{m_{i'}}(\overline{g}(i,j) - \overline{g}(i',j'))^2 r_i w_{i,j}\overline{P}((i,j),(i',j'))}_{B} \tag{47}$$

$$A \leq \sum_{i=1}^{L} \sum_{j=1}^{m_i} \sum_{j'=1}^{n} (\overline{g}(i,j) - \overline{g}(i,j'))^2 r_i w_{i,j} \frac{w_{i,j'}}{\chi^2(p_{i,j} \| p_{i,j'})} \tag{48}$$

$$\leq \sum_{i=1}^{L} \sum_{j=1}^{m_i} r_i w_{i,j} \operatorname{Var}_{p_{i,j}}(g_i) \qquad \text{by Lemma L.1} \tag{49}$$

$$\leq \sum_{i=1}^{L} r_i C \mathscr{E}_i(g_i, g_i). \tag{50}$$

For the second term, we use Lemma L.3.

$$B \leq \sum_{i=1}^{L} \sum_{j=1}^{m_i} \sum_{\substack{i' = i \pm 1 \\ 1 \leq i' \leq L}} \sum_{j'=1}^{m_{i'}} \left[ 6 \operatorname{Var}_{p_{i,j}}(g_i) \chi^2(p_{i,j} \| p_{i',j'}) + 3 \int_{\Omega} (g_i - g_{i'})^2 \min\{p_{i,j}, p_{i',j'}\} \right] r_i w_{i,j} \overline{P}((i,j),(i',j')) \tag{51}$$

$$\leq \sum_{i=1}^{L} \sum_{j=1}^{m_i} \sum_{\substack{i' = i \pm 1 \\ 1 \leq i' \leq L}} \sum_{j'=1}^{m_{i'}} \left[ 6 r_i w_{i,j} \operatorname{Var}_{p_{i,j}}(g_i) w_{i',j'} + 3 r_i w_{i,j} \min\left\{ \frac{r_{i'} w_{i',j'}}{r_i w_{i,j}}, 1 \right\} \int_{\Omega} (g_i - g_{i'})^2 \min\{p_{i,j}, p_{i',j'}\} \right] \tag{52}$$

$$= 12 \sum_{i=1}^{L} \sum_{j=1}^{m_i} r_i w_{i,j} \operatorname{Var}_{p_{i,j}}(g_i) + 3 \sum_{i=1}^{L} \sum_{j=1}^{m_i} \sum_{\substack{i' = i \pm 1 \\ 1 \leq i' \leq L}} \sum_{j=1}^{m_{i'}} \int_{\Omega} (g_i - g_{i'})^2 \min\{r_i w_{i,j} p_{i,j}, r_{i'} w_{i',j'} p_{i',j'}\} \tag{53}$$

$$\leq 12 \sum_{i=1}^{L} r_i C \mathscr{E}_i(g_i, g_i) + 3 \sum_{i=1}^{L} \sum_{\substack{1 \leq i' \leq m \\ i' = i \pm 1}} \int_{\Omega} (g_i - g_{i'})^2 \min\{r_i p_i, r_{i'} p_{i'}\} \tag{54}$$

Then

$$(45) \leq C \sum_{i=1}^{L} r_i \mathscr{E}_i(g_i, g_i) + \frac{\overline{C}}{2} \left( 13 C \sum_{i=1}^{L} r_i \mathscr{E}_i(g_i, g_i) + \frac{12}{\lambda} \frac{\lambda}{2} \sum_{i=1}^{L} \sum_{\substack{1 \leq i' \leq L \\ i' = i \pm 1}} (g_i - g_{i'})^2 \min\{r_i p_i, r_{i'} p_{i'}\} \right) \tag{55}$$

$$\leq \max\left\{ C \left( 1 + \frac{13}{2} \overline{C} \right), \frac{6\overline{C}}{\lambda} \right\} \mathscr{E}(g, g). \tag{56}$$

$\square$

# E  Simulated tempering for gaussians with equal variance

## E.1  Mixtures of gaussians all the way down

**Theorem E.1.** *Let $M$ be the continuous simulated tempering chain for the distributions*

$$p_i \propto \sum_{j=1}^{m} w_j e^{-\beta_i \frac{\|x - \mu_j\|^2}{2\sigma^2}} \tag{57}$$

*with rate* $\Omega\left(\frac{1}{D^2}\right)$, *relative probabilities* $r_i$, *and temperatures* $0 < \beta_1 < \cdots < \beta_L = 1$ *where*

$$D = \max\{\max_j \|\mu_j\|, \sigma\} \tag{58}$$

$$\beta_1 = \Theta\left(\frac{\sigma^2}{D^2}\right) \tag{59}$$

$$\frac{\beta_{i+1}}{\beta_i} \leq 1 + \frac{1}{d} \tag{60}$$

$$L = \Theta\left(d\ln\left(\frac{D}{\sigma}\right) + 1\right) \tag{61}$$

$$r = \frac{\min_i r_i}{\max_i r_i}. \tag{62}$$

*Then* $M$ *satisfies the Poincaré inequality*

$$\mathrm{Var}(g) \leq O\left(\frac{L^2 D^2}{r^2}\right)\mathscr{E}(g,g) = O\left(\frac{\left(d\ln\left(\frac{D}{\sigma}\right)+1\right)^2 D^2}{r^2}\right)\mathscr{E}(g,g). \tag{63}$$

*Proof of Theorem E.1.* Note that forcing $D \leq \sigma$ ensures $\beta_1 = \Omega(1)$. We check all conditions for Theorem D.2.

1. Consider the decomposition where

$$p_{i,j} \propto \exp\left(-\beta_i\frac{\|x - \mu_j\|^2}{2\sigma^2}\right), \tag{64}$$

   $w_{i,j} = w_j$, and and $M_{i,j}$ is the Langevin chain on $p_{i,j}$, so that $\mathscr{E}_{ij}(g_i, g_i) = \int_{\mathbb{R}^d}\|\nabla g_i\|^2 p_{i,j}$. We check (36):

$$\mathscr{E}_i(g_i, g_i) = \int_{\mathbb{R}^d}\|\nabla g_i\|^2 p_i = \int_{\mathbb{R}^d}\|\nabla g_i\|^2\sum_{j=1}^m w_j p_j = \sum_{j=1}^m w_j\mathscr{E}_{i,j}(g_i, g_i). \tag{65}$$

2. By Theorem B.3 and the fact that $\beta_1 = \Omega\left(\frac{\sigma^2}{D^2}\right)$, $\mathscr{E}_{i,j}$ satisfies the Poincaré inequality

$$\mathrm{Var}_{p_{i,j}}(g_i) \leq \frac{\sigma^2}{\beta_i}\mathscr{E}_{i,j}(g_i, g_i) = O(D^2)\mathscr{E}_{i,j}(g_i, g_i). \tag{66}$$

3. To prove a Poincaré inequality for the projected chain, we use the method of canonical paths, Theorem B.4. Consider the graph $G$ on $\bigcup_{i=1}^L\{i\}\times[m_i]$ that is the complete graph on the slice $i = 1$, and the only other edges are vertical edges $(i, j), (i \pm 1, j)$. All the paths we will consider are paths in $G$. For vertices $x = (i, j)$ and $y = (i', j')$, define the canonical path as follows.

   (a) If $j = j'$, without loss of generality $i < i'$. Define the path to be $(i, j), (i+1, j), \ldots, (i', j)$.
   (b) Else, define the path to be $(i, j), (i - 1, j), \ldots, (1, j), (1, j'), \ldots, (i, j')$.

   We calculate the transition probabilities (39), which are given in terms of $\chi^2$ distances. Suppose that $\frac{\beta_{i+1}}{\beta_i} = 1 + \delta$ where $\delta \leq \frac{1}{d}$. Then applying Lemma L.6 to $\Sigma_1 = \beta_i^{-1}I_d$ and

$$\Sigma_2 = \beta_{i+1}^{-1} I_d,$$

$$\chi^2(p_{i,j}||p_{i+1,j}) = \chi^2(N(\mu_j, \beta_i I_d)||N(\mu_j, \beta_{i+1})) \tag{67}$$

$$= \left(\frac{\beta_{i+1}^2}{\beta_i}\right)^{\frac{d}{2}} (2\beta_{i+1} - \beta_i)^{-\frac{d}{2}} - 1 \tag{68}$$

$$= \left(\frac{\beta_{i+1}}{\beta_i}\right)^{\frac{d}{2}} \left(2 - \frac{\beta_i}{\beta_{i+1}}\right)^{-\frac{d}{2}} - 1 \tag{69}$$

$$= O\left((1 + d\delta)\left(2 - \left(\frac{1}{1+\delta}\right)\right)^{-\frac{d}{2}} - 1\right) = O(d\delta) = O(1) \tag{70}$$

Similarly, $\chi^2(p_{i,j}||p_{i-1,j}) = O(1)$. By Lemma L.6 with $\Sigma_1 = \Sigma_2 = \beta_1^{-1} I_d$,

$$\chi^2(p_{1,j}||p_{1,j'}) = \chi^2(N(\mu_j, \beta_1 I_d)||N(\mu_{j'}, \beta_1 I_d)) \tag{71}$$

$$= e^{\beta_1 ||\mu_1 - \mu_2||^2 / \sigma^2} = O(1) \tag{72}$$

when $\beta_1 = O\left(\frac{\sigma^2}{D^2}\right)$.

Note that $|\gamma_{x,y}| \le 2L - 1$. Consider two kinds of edges in $G$.

(a) $z = (i, j)$, $w = (i - 1, j)$. We have

$$\frac{\sum_{\gamma_{x,y} \ni ((i,j),(i-1,j))} |\gamma_{x,y}| p(x) p_{st}(y)}{p((i,j)) P((i,j),(i-1,j))} \le \frac{(2L-1)p(S)p(S^c)}{p((i,j)) P((i,j),(i-1,j))} \tag{73}$$

where $S = \{i, \ldots, L\} \times \{j\}$. This follows because cutting the edge $zw$ splits the graph into 2 connected components, one of which is $S$; the paths which go through $zw$ are exactly those between $x, y$ where one of $x, y$ is a subset of $S$ and the other is not. Now note

$$\frac{p(S)}{p((i,j))} = \frac{p(\{i, \ldots L\} \times \{j\})}{p((i,j))} \le \frac{L}{r} \tag{74}$$

$$p(S^c) \le 1 \tag{75}$$

$$P((i,j),(i-1,j)) = \frac{\min\left\{\frac{r_{i'} w_{i',j'}}{r_i w_{i,j}}, 1\right\}}{\max\{\chi^2(p_{i,j}||p_{i',j'}), \chi^2(p_{i',j'}||p_{i,j})\}} = \Omega(r) \tag{76}$$

by (70) and (39) so $(73) = O\left(\frac{L^2}{r^2}\right)$.

(b) $z = (1, j)$, $w = (1, k)$. We have

$$\frac{\sum_{\gamma_{x,y} \ni ((1,j),(1,k))} |\gamma_{x,y}| p(x) p(y)}{p((1,j)) P((1,j),(1,k))} \le \frac{(2L-1)p_{st}([L] \times \{j\}) p((i,j))}{p((i,j)) P((1,j),(1,k))}. \tag{77}$$

because the paths going through $zw$ are exactly those between $[L] \times \{j\}$ and $[L] \times \{k\}$. Now note

$$\frac{p([L] \times \{j\})}{p((1,j))} \le \frac{L}{r} \tag{78}$$

$$p([L] \times \{k\}) = w_k \tag{79}$$

$$P((1,j),(1,k)) = \frac{w_k}{\max\{\chi^2(p_{1,j}||p_{1,k}), \chi^2(p_{1,j}||p_{1,k})\}} = \Omega(w_k). \tag{80}$$

Thus $(77) = O\left(\frac{L^2}{r}\right)$.

By Theorem B.4, the projected chain satisfies a Poincaré inequality with constant $O\left(\frac{L^2}{r^2}\right)$.

Thus by Theorem D.2, the simulated tempering chain satisfies a Poincaré inequality with constant

$$O\left(\max\left\{D^2\left(1+\frac{L^2}{r^2}\right), \frac{L^2}{r^2\lambda}\right\}\right).$$

(81)

Taking $\lambda = \frac{1}{D^2}$ makes this $O\left(\frac{D^2 L^2}{r^2}\right)$. $\qquad\square$

**Remark E.2.** *Note there is no dependence on either $w_{\min}$ or the number of components.*

*If $p \propto \sum_{j=1}^m w_j e^{-\frac{\|x-\mu_j\|^2}{2\sigma^2}}$ and we have access to $\nabla \ln(p * N(0, \tau I))$ for any $\tau$, then we can sample from $p$ efficiently, no matter how many components there are. In fact, passing to the continuous limit, we can sample from any $p$ in the form $p = w * N(0, \sigma^2 I_d)$ where $\|w\|_1 = 1$ and $\mathrm{Supp}(w) \subseteq B_D$.*

*In this way, Theorem E.1 says that evolution of $p$ under the heat kernel is the most "natural" way to do simulated tempering. We don't have access to $p * N(0, \tau I)$, but we will show that $p^\beta$ approximates it well (within a factor of $\frac{1}{w_{\min}}$).*

*Entropy-SGD [CCSL16] attempts to estimate $\nabla \ln(p * N(0, \tau I))$ for use in a temperature-based algorithm; this remark provides some heuristic justification for why this is a natural choice.*

## E.2 Comparing to the actual chain

The following lemma shows that changing the temperature is approximately the same as changing the variance of the gaussian. We state it more generally, for arbitrary mixtures of distributions in the form $e^{-f_i(x)}$.

**Lemma E.3** (Approximately scaling the temperature). *Let $p_i(x) = e^{-f_i(x)}$ be probability distributions on $\Omega$ such that for all $\beta > 0$, $\int_\Omega e^{-\beta f_i(x)}\, dx < \infty$. Let*

$$p(x) = \sum_{i=1}^n w_i p_i(x)$$

(82)

$$f(x) = -\ln p(x)$$

(83)

*where $w_1, \ldots, w_n > 0$ and $\sum_{i=1}^n w_i = 1$. Let $w_{\min} = \min_{1 \le i \le n} w_i$.*

*Define the distribution at inverse temperature $\beta$ to be $p_\beta(x)$, where*

$$g_\beta(x) = e^{-\beta f(x)}$$

(84)

$$Z_\beta = \int_\Omega e^{-\beta f(x)}\, dx$$

(85)

$$p_\beta(x) = \frac{g_\beta(x)}{Z_\beta}.$$

(86)

*Define the distribution $\widetilde{p}_\beta(x)$ by*

$$\widetilde{g}_\beta(x) = \sum_{i=1}^n w_i e^{-\beta f_i(x)}$$

(87)

$$\widetilde{Z}_\beta = \int_\Omega \sum_{i=1}^n w_i e^{-\beta f_i(x)}\, dx$$

(88)

$$\widetilde{p}_\beta(x) = \frac{\widetilde{g}_\beta(x)}{\widetilde{Z}_\beta}.$$

(89)

*Then for $0 \le \beta \le 1$ and all $x$,*

$$g_\beta(x) \in \left[1, \frac{1}{w_{\min}}\right] \widetilde{g}_\beta$$

(90)

$$p_\beta(x) \in \left[1, \frac{1}{w_{\min}}\right] \widetilde{p}_\beta \frac{\widetilde{Z}_\beta}{Z_\beta} \subset \left[w_{\min}, \frac{1}{w_{\min}}\right] \widetilde{p}_\beta.$$

(91)

*Proof.* By the Power-Mean inequality,

$$g_\beta(x) = \left( \sum_{i=1}^n w_i e^{-f_i(x)} \right)^\beta \tag{92}$$

$$\geq \sum_{i=1}^n w_i e^{-\beta f_i(x)} = \widetilde{g}_\beta(x). \tag{93}$$

On the other hand, given $x$, setting $j = \operatorname{argmin}_i f_i(x)$,

$$g_\beta(x) = \left( \sum_{i=1}^n w_i e^{-f_i(x)} \right)^\beta \tag{94}$$

$$\leq (e^{-f_j(x)})^\beta \tag{95}$$

$$\leq \frac{1}{w_{\min}} \sum_{i=1}^n w_i e^{-\beta f_i(x)} = \frac{1}{w_{\min}} \widetilde{g}_\beta(x). \tag{96}$$

This gives (90). This implies $\frac{\widetilde{Z}_\beta}{Z_\beta} \in [w_{\min}, 1]$, which gives (91). $\qquad\square$

**Lemma E.4.** *Suppose* $\|f_1 - f_2\|_\infty \leq \frac{\Delta}{2}$ *and* $p_1 \propto e^{-f_1}$, $p_2 \propto e^{-f_2}$ *are probability distributions on* $\mathbb{R}^d$. *Then*

$$\frac{\mathscr{E}_{p_1}(g)}{\|g\|_{p_1}^2} \geq e^{-2\Delta} \frac{\mathscr{E}_{p_2}(g)}{\|g\|_{p_2}^2}. \tag{97}$$

*Proof.* The ratio between $p_1$ and $p_2$ is at most $e^\Delta$, so

$$\frac{\int_{\mathbb{R}^d} \|\nabla g\|^2 p_1 \, dx}{\int_{\mathbb{R}^d} \|g\|^2 p_1 \, dx} \geq \frac{e^{-\Delta} \int_{\mathbb{R}^d} \|\nabla g\|^2 p_2 \, dx}{e^\Delta \int_{\mathbb{R}^d} \|g\|^2 p_2 \, dx}. \tag{98}$$

$\qquad\square$

**Lemma E.5.** *Let* $M$ *and* $\widetilde{M}$ *be two continuous simulated tempering Langevin chains with functions* $f_i$, $\widetilde{f}_i$,, *respectively, for* $i \in [L]$, *with rate* $\lambda$, *and with relative probabilities* $r_i$. *Let their Dirichlet forms be* $\mathscr{E}$ *and* $\widetilde{\mathscr{E}}$ *and their stationary distributions be* $p$ *and* $\widetilde{p}$.

*Suppose that* $\left\| f_i(x) - \widetilde{f}_i(x) \right\|_\infty \leq \frac{\Delta}{2}$. *Then*

$$\frac{\mathscr{E}(g,g)}{\operatorname{Var}_p(g)} \geq e^{-2\Delta} \frac{\widetilde{\mathscr{E}}(g,g)}{\operatorname{Var}_{\widetilde{p}}(g)}. \tag{99}$$

*Proof.* By Lemma E.4,

$$\frac{\sum_{i=1}^L \mathscr{E}_i(g_i, g_i)}{\operatorname{Var}_{p_i}(g_i)} \geq e^{-2\Delta} \frac{\sum_{i=1}^L \widetilde{\mathscr{E}}_i(g_i, g_i)}{\operatorname{Var}_{\widetilde{p}_i}(g_i)} \tag{100}$$

$$\implies \frac{\sum_{i=1}^L r_i \mathscr{E}_i(g_i, g_i)}{\operatorname{Var}_p(g_i)} \geq e^{-2\Delta} \frac{\sum_{i=1}^L r_i \widetilde{\mathscr{E}}_i(g_i, g_i)}{\operatorname{Var}_{\widetilde{p}}(g_i)}. \tag{101}$$

By Lemma E.3, we have $\frac{p_i}{\widetilde{p}_i} \in [A, Ae^\Delta]$, $\frac{p_j}{\widetilde{p}_j} \in [B, Be^\Delta]$ for some $A, B$, so $\frac{\min\{r_ip_i, r_jp_j\}}{\min\{r_i\widetilde{p}_i, r_j\widetilde{p}_j\}} \in [C, Ce^\Delta]$ for some $C$. Hence, for some $D, E$,

$$\frac{\int(g_i - g_j)^2 \min\{r_ip_i, r_jp_j\}}{\int(g_i - g_j)^2 \min\{r_i\widetilde{p}_i, r_j\widetilde{p}_j\}} \in [D, De^\Delta] \tag{102}$$

$$\frac{\mathrm{Var}_{p_i}(g_i)}{\mathrm{Var}_{\widetilde{p}_i}(\widetilde{g}_i)} \in [E, Ee^\Delta] \tag{103}$$

$$\implies \frac{\frac{\lambda}{4}\sum_{j=i\pm1}\int(g_i - g_j)^2 \min\{r_ip_i, r_jp_j\}}{r_i \mathrm{Var}_{p_i}(g_i)} \geq e^{-2\Delta}\frac{\frac{\lambda}{4}\sum_{j=i\pm1}\int(g_i - g_j)^2 \min\{r_i\widetilde{p}_i, r_j\widetilde{p}_j\}}{r_j \mathrm{Var}_{p_j}(g)} \tag{104}$$

$$\frac{\frac{\lambda}{4}\sum_{i=1}^L\sum_{j=i\pm1}\int(g_i - g_j)^2 \min\{r_ip_i, r_jp_j\}}{\mathrm{Var}_p(g)} \geq e^{-2\Delta}\frac{\frac{\lambda}{4}\sum_{i=1}^L\sum_{j=i\pm1}\int(g_i - g_j)^2 \min\{r_i\widetilde{p}_i, r_j\widetilde{p}_j\}}{\mathrm{Var}_{\widetilde{p}}(g)} \tag{105}$$

Adding (101) and (105) gives the result. $\square$

**Theorem E.6.** *Suppose* $\sum_{j=1}^m w_j = 1$, $w_{\min} = \min_{1\leq j\leq m} w_i > 0$, *and* $D = \max\{\max_{1\leq j\leq m}\|\mu_j\|, \sigma\}$. *Let $M$ be the continuous simulated tempering chain for the distributions*

$$p_i \propto \left(\sum_{j=1}^m w_j e^{-\frac{\|x-\mu_j\|^2}{2\sigma^2}}\right)^{\beta_i} \tag{106}$$

*with rate* $O\left(\frac{1}{D^2}\right)$, *relative probabilities* $r_i$, *and temperatures* $0 < \beta_1 < \cdots < \beta_L = 1$ *satisfying the same conditions as in Theorem E.1. Then $M$ satisfies the Poincaré inequality*

$$\mathrm{Var}(g) \leq O\left(\frac{L^2 D^2}{r^2 w_{\min}^2}\right)\mathscr{E}(g, g) = O\left(\frac{d^2\left(\ln\left(\frac{D}{\sigma}\right)\right)^2 D^2}{r^2 w_{\min}^2}\right)\mathscr{E}(g, g). \tag{107}$$

*Proof.* Let $\widetilde{p}_i$ be the probability distributions in Theorem E.1 with the same parameters as $p_i$ and let $\widetilde{p}$ be the stationary distribution of that simulated tempering chain. By Theorem E.1, $\mathrm{Var}_{\widetilde{p}}(g) = O\left(\frac{L^2 D^2}{r^2}\right)\mathscr{E}\widetilde{p}(g, g)$. Now use By Lemma E.3, $\frac{p_i}{\widetilde{p}_i} \in \left[1, \frac{1}{w_{\min}}\right]\frac{\widetilde{Z}_i}{Z_i}$. Now use Lemma E.5 with $e^\Delta = \frac{1}{w_{\min}}$. $\square$

## F  Discretization

**Lemma F.1.** *Fix times* $0 < T_1 < \cdots < T_n \leq T$.

*Let* $p^T, q^T : [L] \times \mathbb{R}^d \to \mathbb{R}$ *be defined as follows.*

1. $p^T$ *is the continuous simulated tempering Markov as in Definition C.1 but with fixed transition times* $T_1, \ldots, T_n$. *The component chains are Langevin diffusions on* $p_i \propto \left(\sum_{j=1}^m w_j e^{-f_0(x-\mu_i)}\right)^{\beta_i}$.

2. $q^T$ *is the discretized version as in Algorithm (1), again with fixed transition times* $T_1, \ldots, T_n$, *and with step size* $\eta \leq \frac{\sigma^2}{2}$.

*Then*

$$KL(p^T \| q^T) \lesssim \eta^2 D^6 K^7 \left(D^2\frac{K^2}{\kappa} + d\right)Tn + \eta^2 D^3 K^3 \max_i \mathbb{E}_{x\sim p^0(\cdot, i)}\|x - x^*\|_2^2 + \eta D^2 K^2 dT$$

*where $x^*$ is the maximum of* $\sum_{j=1}^m w_j e^{-f_0(x-\mu_j)}$ *and satisfies* $\|x^*\| = O(D)$ *where* $D = \max\|\mu_j\|$.

Before proving the above statement, we make a note on the location of $x^*$ to make sense of $\max_i \mathbb{E}_{x \sim p^0(\cdot, i)} \|x - x^*\|_2^2$. Namely, we show:

**Lemma F.2** (Location of minimum). *Let $x^* = argmin_{x \in \mathbb{R}^d} f(x)$. Then, $\|x^*\| \leq D\sqrt{\frac{K}{\kappa} + 1}$.*

*Proof.* Recall that $f_i(x) = f_0(x - \mu_i)$. We claim that $f(0) \leq \frac{1}{2}KD^2$. Indeed, by smoothness, we have $f_i(0) \leq \frac{1}{2}K\|\mu_i\|^2$, which implies that $f(0) \leq \frac{1}{2}KD^2$.

Hence, it follows that $\min_{x \in \mathbb{R}^d} f(x) \leq \frac{1}{2}KD^2$. However, for any $x$, it holds that

$$
\begin{aligned}
f(x) &\geq \frac{1}{2} \min_i \kappa \|\mu_i - x\|^2 \\
&\geq \frac{1}{2}\kappa \left( \|x\|^2 - \max_i \|\mu_i\|^2 \right) \\
&\geq \frac{1}{2}\kappa \left( \|x\|^2 - D^2 \right)
\end{aligned}
$$

Hence, if $\|x\| > D\sqrt{\frac{K}{\kappa} + 1}$, $f(x) > \min_{x \in \mathbb{R}^d} f(x)$. This implies the statement of the lemma. $\qquad\square$

We prove a few technical lemmas. First, we prove that the continuous chain is essentially contained in a ball of radius $D$. More precisely, we show:

**Lemma F.3** (Reach of continuous chain). *Let $P_T^\beta(X)$ be the Markov kernel corresponding to evolving Langevin diffusion*

$$
\frac{dX_t}{dt} = -\beta \nabla f(X_t) + dB_t
$$

*with $\tilde{f}$ and $D$ are as defined in 1 for time $T$. Then,*

$$
\mathbb{E}[\|X_t - x^*\|^2] \leq \mathbb{E}[\|X_0 - x^*\|^2] + \left( 400\beta \frac{D^2 K^2}{\kappa} + 2d \right) T
$$

*Proof.* Let $Y_t = \|X_t - x^*\|^2$. By Itôs Lemma, we have

$$
dY_t = -2 \left\langle X_t - x^*, \beta \sum_{i=1}^m \frac{w_i e^{-f_i(X_t)} \nabla f_i(X_t)}{\sum_{j=1}^m w_j e^{-f_j(X_t)}} \right\rangle + 2d\, dt + \sqrt{8} \sum_{i=1}^d (X_t)_i\, d(B_i)_t \qquad (108)
$$

We will show that

$$
-\langle X_t - x^*, \nabla f_i(X_t) \rangle \leq 100 \frac{D^2 K^2}{\kappa}
$$

Indeed, since $f_i(x) = f_0(x - \mu_i)$, by (108), we have

$$
\langle X_t, \nabla f_i(X_t) \rangle \geq \frac{\kappa}{2}\|X_t\|^2 - \frac{D^2(2\kappa + K)^2}{2\kappa} - KD^2
$$

Also, by the Hessian bound $\kappa I \preceq \nabla^2 f_0(x) \preceq KI$, we have

$$
\langle x^*, \nabla f_i(X_t) \rangle \leq \|x^*\| \|\nabla f_i(X_t)\| \leq D\sqrt{\frac{K}{\kappa} + 1}\|X_t - \mu_i\| \leq D\sqrt{\frac{K}{\kappa} + 1}(\|X_t\| + D)
$$

Hence,

$$
-\langle X_t - x^*, \nabla f_i(X_t) \rangle \leq -\frac{\kappa}{2}\|X_t\|^2 - \frac{D^2(2\kappa + K)^2}{2\kappa} - D\sqrt{\frac{K}{\kappa} + 1}(\|X_t\| + D)
$$

Solving for the extremal values of the quadratic on the RHS, we get

$$
-\langle X_t - x^*, \nabla f_i(X_t) \rangle \leq 100 \frac{D^2 K^2}{\kappa}
$$

Together with (108), we get

$$dY_t \leq 100\beta \frac{D^2 K^2}{\kappa} + 2d\, dt + \sqrt{8} \sum_{i=1}^{d} (X_t)_i \, d(B_i)_t$$

Integrating, we get

$$Y_t \leq Y_0 + 400\beta \frac{D^2 K^2}{\kappa} T + 2dT + \sqrt{8} \int_0^T \sum_{i=1}^{d} (X_t)_i \, d(B_i)_t$$

Taking expectations and using the martingale property of the Itô integral, we get the claim of the lemma. $\qquad\square$

Next, we prove a few technical bound the drift of the discretized chain after $T/\eta$ discrete steps. The proofs follow similar calculations as those in [Dal16].

We will first need to bound the Hessian of $\tilde{f}$.

**Lemma F.4** (Hessian bound).

$$-2(DK)^2 I \preceq \nabla^2 f(x) \preceq KI, \forall x \in \mathbb{R}^d$$

*Proof.* For notational convenience, let $p(x) = \sum_{i=1}^{m} w_i e^{-f_i(x)}$. Note that $f(x) = -\log p(x)$. We proceed to the upper bound first. The Hessian of $f$ satisfies

$$\nabla^2 f = \frac{\sum_i w_i e^{-f_i} \nabla^2 f_i}{p} - \frac{\frac{1}{2}\sum_{i,j} w_i w_j e^{-f_i} e^{-f_j} (\nabla f_i - \nabla f_j)^{\otimes 2}}{p^2}$$
$$\preceq \max_i \nabla^2 f_i \preceq KI$$

as we need. As for the lower bound, we have

$$\nabla^2 f \succeq -\frac{1}{2} \sum_{i,j} \|\nabla f_i - \nabla f_j\|^2$$
$$\succeq -\max_{i,j} e^{-f_i} e^{-f_j} (\nabla f_i - \nabla f_j)^{\otimes 2}$$

But notice that since $f_i(x) = f_0(x + \mu_i)$, we have

$$\|\nabla f_i(x) - \nabla f_j(x)\| = \|\nabla f_0(x + \mu_i) - \nabla f_0(x + \mu_j)\|$$
$$\leq K\|\mu_i - \mu_j\|$$
$$\leq 2DK$$

where the next-to-last inequality follows from the strong-convexity of $f_0$. This proves the statement of the lemma.

$\qquad\square$

We introduce the following piece of notation in the following portion: we denote by $P_T(x, i) : \mathbb{R}^d \times [L] \to \mathbb{R}, \forall x \in \mathbb{R}^d, i \in [L]$ the distribution on $\mathbb{R}^d \times [L]$ corresponding to running the Langevin diffusion chain for $T$ time steps on the first coordinate, starting at $x \times \{i\}$, and keeping the second coordinates fixed. Let us define by $\widehat{P_T}(x, i) : \mathbb{R}^d \times [L] \to \mathbb{R}$ the analogous distribution, except running the discretized Langevin diffusion chain for $\frac{T}{\eta}$ time steps on the first coordinate, for $\frac{T}{\eta}$ an integer.

**Lemma F.5** (Bounding interval drift). *In the setting of this section, let $x \in \mathbb{R}^d, i \in [L]$, and let $\eta \leq \frac{1}{K}$.*

$$KL(P_T(x,i)||\widehat{P_T}(x,i)) \leq \frac{4D^6 \eta^2 K^7}{3} \left( \|x - x^*\|_2^2 + 8Td \right) + dTD^2 \eta K^2$$

*Proof.* Let $x_j, i \in [0, T/\eta - 1]$ be a random variable distributed as $\widehat{P_{\eta j}}(x, i)$. By Lemma 2 in [Dal16] and Lemma F.4 , we have

$$\mathrm{KL}(P_T(x,i)||\widehat{P_T}(x,i)) \leq \frac{\eta^3 D^2 K^2}{3} \sum_{k=0}^{T/\eta-1} \mathbb{E}[\|\nabla f(x^k)\|_2^2] + dT\eta D^2 K^2$$

Similarly, the proof of Corollary 4 in [Dal16] implies that

$$\eta \sum_{k=0}^{T/\eta-1} \mathbb{E}[\|\nabla f(x^k)\|_2^2] \leq 4D^4 K^4 \|x - x^*\|_2^2 + 8DKTd$$

Plugging this in, we get the statement of the lemma. □

Finally, we prove a convenient decomposition theorem for the KL divergence of two mixtures of distributions, in terms of the KL divergence of the weights and the components in the mixture. Concretely:

**Lemma F.6.** *Let $w, w' : I \to \mathbb{R}$ be distributions over a domain $I$ with full support. Let $p_i, q_i : \forall i \in I$ be distributions over an arbitrary domain. Then:*

$$KL\left(\int_{i\in I} w_i p_i || \int_{i\in I} w_i' q_i\right) \leq KL(w||w') + \int_{i\in I} w_i KL(p_i||q_i)$$

*Proof.* Overloading notation, we will use $KL(a||b)$ for two measures $a, b$ even if they are not necessarily probability distributions, with the obvious definition.

$$
\begin{aligned}
\mathrm{KL}\left(\int_{i\in I} w_i p_i || \int_{i\in I} w_i' q_i\right) &= \mathrm{KL}\left(\int_{i\in I} w_i p_i || \int_{i\in I} w_i q_i \frac{w_i'}{w_i}\right) \\
&\leq \int_{i\in I} w_i \mathrm{KL}\left(p_i || q_i \frac{w_i'}{w_i}\right) \\
&= \int_{i\in I} w_i \log\left(\frac{w_i}{w_i'}\right) + \mathrm{KL}(p_i||q_i) \\
&= \mathrm{KL}(w||w') + \int_{i\in I} w_i \mathrm{KL}(p_i||q_i)
\end{aligned}
$$

where the first inequality holds due to the convexity of KL divergence.

□

With this in mind, we can prove the main claim:

*Proof of F.1.* Let's denote by $R(x, i) : \mathbb{R}^d \times [L] \to \mathbb{R}$ the distribution on $\mathbb{R}^d \times [L]$, running the Markov transition matrix corresponding to a Type 2 transition in the simulated tempering chain, starting at $(x, i)$.

We will proceed by induction. Towards that, we can obviously write

$$p^{T_{i+1}} = \frac{1}{2}\left(\int_{x\in\mathbb{R}^d} \sum_{j=0}^{L-1} p^{T_i}(x,j) P_{T_{i+1}-T_i}(x,j)\right) + \frac{1}{2}\left(\int_{x\in\mathbb{R}^d} \sum_{j=0}^{L-1} p^{T_i}(x,j) R(x,j)\right)$$

and similarly

$$q^{T_{i+1}}(x,j) = \frac{1}{2}\left(\int_{x\in\mathbb{R}^d} \sum_{j=0}^{L-1} q^{T_i}(x,j) \widehat{P_{T_{i+1}-T_i}}(x,j)\right) + \frac{1}{2}\left(\int_{x\in\mathbb{R}^d} \sum_{j=0}^{L-1} q^{T_i}(x,j) R(x,j)\right)$$

(Note: the $R$ transition matrix doesn't change in the discretized vs continuous version.)

By convexity of KL divergence, we have

$$\mathrm{KL}(p^{T_{i+1}}||q^{T_{i+1}}) \leq \frac{1}{2}\mathrm{KL}\left(\int_{x\in\mathbb{R}^d}\sum_{j=0}^{L-1}p^{T_i}(x,j)P_{T_{i+1}-T_i}(x,j)||\int_{x\in\mathbb{R}^d}\sum_{j=0}^{L-1}q^{T_i}(x,j)\widehat{P_{T_{i+1}-T_i}}(x,j)\right)$$

$$+\frac{1}{2}\mathrm{KL}\left(\int_{x\in\mathbb{R}^d}\sum_{j=0}^{L-1}p^{T_i}(x,j)R(x,j)||\int_{x\in\mathbb{R}^d}\sum_{j=0}^{L-1}q^{T_i}(x,j)R(x,j)\right)$$

By Lemma F.6, we have that

$$\mathrm{KL}\left(\int_{x\in\mathbb{R}^d}\sum_{j=0}^{L-1}p^{T_i}(x,j)R(x,j)||\int_{x\in\mathbb{R}^d}\sum_{j=0}^{L-1}q^{T_i}(x,j)R(x,j)\right)\leq \mathrm{KL}(p^{T_i}||q^{T_i})$$

Similarly, by Lemma F.5 together with Lemma F.6 we have

$$\mathrm{KL}\left(\int_{x\in\mathbb{R}^d}\sum_{j=0}^{L-1}p^{T_i}(x,j)P_{T_{i+1}-T_i}(x,j)||\int_{x\in\mathbb{R}^d}\sum_{j=0}^{L-1}q^{T_i}(x,j)\widehat{P_{T_{i+1}-T_i}}(x,j)\right)\leq$$

$$\mathrm{KL}(p^{T_i}||q^{T_i})+\frac{4D^6K^6\eta^2}{3}\left(\max_j\mathbb{E}_{x\sim p^{t_i}(\cdot,j)}\|x-x^*\|_2^2+8(T_{i+1}-T_i)d\right)+d(T_{i+1}-T_i)\eta K^2$$

By Lemmas F.3 and F.2, we have that for any $j\in[0,L-1]$,

$$\mathbb{E}_{x\sim p^{T_i}(\cdot,j)}\|x-x^*\|_2^2\leq\mathbb{E}_{x\sim p^{T_{i-1}}(\cdot,j)}\|x\|_2+\left(400\frac{D^2K^2}{\kappa}+2d\right)(T_i-T_{i-1})$$

Hence, inductively, we have $\mathbb{E}_{x\sim p^{T_i}(\cdot,j)}\|x-x^*\|_2^2\leq\mathbb{E}_{x\sim p^0(\cdot,j)}\|x-x^*\|_2^2+\left(400\frac{D^2K^2}{\kappa}+2d\right)T_i$

Putting together, we have

$$\mathrm{KL}(p^{T_{i+1}}||q^{T_{i+1}})\leq\mathrm{KL}(p^{T_i}||q^{T_i})+\frac{4\eta^2D^6K^7}{3}\left(\max_j\mathbb{E}_{x\sim p^0(\cdot,j)}\|x-x^*\|_2^2+\left(400\frac{D^2D^2K^2}{\kappa}+2d\right)T+8Td\right)+dT\eta D^2 I$$

By induction, we hence have

$$\mathrm{KL}(p^T||q^T)\lesssim\eta^2D^6K^7\left(D^2\frac{K^2}{\kappa}+d\right)Tn+\eta^2D^3K^3\max_i\mathbb{E}_{x\sim p^0(\cdot,i)}\|x-x^*\|_2^2+\eta D^2K^2dT$$

as we need. □

# G   Proof of main theorem

Before putting everything together, we show how to estimate the partition functions. We will apply the following to $g_1(x)=e^{-\beta_\ell f(x)}$ and $g_2(x)=e^{-\beta_{\ell+1}f(x)}$.

**Lemma G.1** (Estimating the partition function to within a constant factor). *Suppose that $p_1(x)=\frac{g_1(x)}{Z_1}$ and $p_2(x)=\frac{g_2(x)}{Z_2}$ are probability distributions on $\Omega$. Suppose $\widetilde{p}_1$ is a distribution such that $d_{TV}(\widetilde{p}_1,p_1)<\frac{\varepsilon}{2C^2}$, and $\frac{g_2(x)}{g_1(x)}\in[0,C]$ for all $x\in\Omega$. Given $n$ samples from $\widetilde{p}_1$, define the random variable*

$$\overline{r}=\frac{1}{n}\sum_{i=1}^{n}\frac{g_2(x_i)}{g_1(x_i)}. \tag{109}$$

*Let*

$$r=\mathbb{E}_{x\sim p_1}\frac{g_2(x)}{g_1(x)}=\frac{Z_2}{Z_1} \tag{110}$$

*and suppose $r\geq\frac{1}{C}$. Then with probability $\geq 1-e^{-\frac{n\varepsilon^2}{2C^4}}$,*

$$\left|\frac{\overline{r}}{r}-1\right|\leq\varepsilon. \tag{111}$$

*Proof.* We have that

$$\left| \mathbb{E}_{x\sim\widetilde{p}_1} \frac{g_2(x)}{g_1(x)} - \mathbb{E}_{x\sim p_1} \frac{g_2(x)}{g_1(x)} \right| \leq C d_{TV}(\widetilde{p}_1, p_1) \leq \frac{\varepsilon}{2C}. \tag{112}$$

The Chernoff bound gives

$$\mathbb{P}\left( \left| r - \mathbb{E}_{x\sim\widetilde{p}_1} \frac{g_2(x)}{g_1(x)} \right| \geq \frac{\varepsilon}{2C} \right) \leq e^{-\frac{n\left(\frac{\varepsilon}{2C}\right)^2}{2\left(\frac{C}{2}\right)^2}} = e^{-\frac{n\varepsilon^2}{2C^4}}. \tag{113}$$

Combining (112) and (113) using the triangle inequality,

$$\mathbb{P}\left( |\overline{r} - r| \geq \frac{1}{\varepsilon} C \right) \leq e^{-\frac{n\varepsilon^2}{2C^4}}. \tag{114}$$

Dividing by $r$ and using $r \geq \frac{1}{C}$ gives the result. $\qquad\square$

**Lemma G.2.** *Suppose that Algorithm 1 is run on* $f(x) = -\ln\left(\sum_{j=1}^m w_j \exp\left(-\frac{\|x-\mu_j\|^2}{2\sigma^2}\right)\right)$ *with temperatures* $0 < \beta_1 < \cdots < \beta_\ell \leq 1$, $\ell \leq L$, *rate* $\lambda$, *and with partition function estimates* $\widehat{Z}_1, \ldots, \widehat{Z}_\ell$ *satisfying*

$$\left| \frac{\widehat{Z}_i}{Z_i} \Big/ \frac{\widehat{Z}_1}{Z_1} \right| \in \left[ \left(1 - \frac{1}{L}\right)^{i-1}, \left(1 + \frac{1}{L}\right)^{i-1} \right] \tag{115}$$

*for all* $1 \leq i \leq \ell$. *Suppose* $\sum_{j=1}^m w_j = 1$, $w_{\min} = \min_{1\leq j\leq m} w_i > 0$, *and* $D = \max\{\max_{1\leq j\leq m} \|\mu_j\|, \sigma\}$, *and the parameters satisfy*

$$\lambda = \Theta\left(\frac{1}{D^2}\right) \tag{116}$$

$$\beta_1 = \Theta\left(\frac{\sigma^2}{D^2}\right) \tag{117}$$

$$\frac{\beta_{i+1}}{\beta_i} \leq 1 + \frac{1}{d + \ln\left(\frac{1}{w_{\min}}\right)} \tag{118}$$

$$L = \Theta\left(\left(d + \ln\left(\frac{1}{w_{\min}}\right)\right)\ln\left(\frac{D}{\sigma}\right) + 1\right) \tag{119}$$

$$T = \Omega\left(\frac{L^2 D^2 \ln\left(\frac{\ell}{\varepsilon w_{\min}}\right)}{w_{\min}^2}\right) \tag{120}$$

$$\eta = O\left(\frac{\sigma^3 \varepsilon}{D^2} \min\left\{ \frac{\sigma^4}{\left(\frac{D}{\sigma} + \sqrt{d}\right) T}, \frac{1}{D^{\frac{1}{2}}}, \frac{\sigma\varepsilon}{dT} \right\}\right) \tag{121}$$

*Let* $q^0$ *be the distribution* $\left(N\left(0, \frac{\sigma^2}{\beta_1}\right), 1\right)$ *on* $[\ell] \times \mathbb{R}^d$. *Then the distribution* $q^T$ *after running time* $T$ *satisfies* $\|p - q^T\|_1 \leq \varepsilon$.

*Setting* $\varepsilon = O\left(\frac{1}{\ell L}\right)$ *above and taking* $n = \Omega\left(L^2 \ln\left(\frac{1}{\delta}\right)\right)$ *samples, with probability* $1 - \delta$ *the estimate*

$$\widehat{Z}_{\ell+1} = \overline{r}\widehat{Z}_\ell, \qquad\qquad \overline{r} := \frac{1}{n}\sum_{j=1}^n e^{(-\beta_{\ell+1}+\beta_\ell)f_i(x_j)} \tag{122}$$

*also satisfies* (115).

*Proof.* By the triangle inequality,

$$\|p - q^T\|_1 \leq \|p - p^T\|_1 + \|p^T - q^T\|_1. \tag{123}$$

For the first term, by Cauchy-Schwarz,

$$\left\|p - p^T\right\|_1 \le \sqrt{\chi^2(p\|p^T)} \le e^{-\frac{T}{2C}}\sqrt{\chi^2(p\|p^0)} \tag{124}$$

where $C = O\left(\frac{d^2\left(\ln\left(\frac{D}{\sigma}\right)\right)^2 D^2}{w_{\min}^2}\right)$ is an upper bound on the Poincaré constant as in Theorem E.6. (The assumption on $\widehat{Z}_i$ means that $r \le e$.) Let $p_i$ be the distribution of $p$ on the $i$th temperature, and $\widetilde{p}_i$ be as in Lemma E.3.

To calculate $\chi^2(p\|p^0)$, first note by Lemma L.6, the $\chi^2$ distance between $N\left(0, \frac{\sigma^2}{\beta_1}I_d\right)$ and $N(\mu, \frac{\sigma^2}{\beta_1}I_d)$ is $\le e^{\|\mu\|^2\beta_1/\sigma^2}$. Then

$$\chi^2(p\|p^0) \tag{125}$$

$$= O(\ell)\chi^2\left(p_1\|N\left(0, \frac{\sigma^2}{\beta_1}I_d\right)\right) \tag{126}$$

$$= O\left(\frac{\ell}{w_{\min}}\right)\left(1 + \chi^2\left(\widetilde{p}_1\|N\left(0, \frac{\sigma^2}{\beta_1}I_d\right)\right)\right) \qquad \text{by Lemma E.3 and Lemma L.5} \tag{127}$$

$$= O\left(\frac{\ell}{w_{\min}}\right)\left(1 + \sum_{j=1}^{m} w_j\chi^2\left(N\left(\mu_j, \frac{\sigma^2}{\beta_1}I_d\right)\|N\left(0, \frac{\sigma^2}{\beta_1}I_d\right)\right)\right) \qquad \text{by Lemma L.4} \tag{128}$$

$$= O\left(\frac{e^{\frac{D^2\beta_1}{\sigma^2}}\ell}{w_{\min}}\right) = O\left(\frac{\ell}{w_{\min}}\right). \tag{129}$$

Together with (124) this gives $\left\|p - p^T\right\|_1 \le \frac{\varepsilon}{3}$.

For the second term $\left\|p^T - q^T\right\|_1$, we first condition on there not being too many transitions before time $T$. Let $N_T = \max\{n : T_n \le T\}$ be the number of transitions. Let $C$ be as in Lemma L.16. Note that $\left(\frac{Cn}{T\lambda}\right)^{-n} \le \varepsilon \iff e^{n\left(\frac{T\lambda}{C}\right) - \ln n} \le \varepsilon$, and that this inequality holds when $n \ge \frac{eT\lambda}{C} + \ln\left(\frac{1}{\varepsilon}\right)$. We have by Lemma L.16 that $\mathbb{P}(N_T \ge \frac{eT\lambda}{C} + \ln\left(\frac{1}{\varepsilon}\right)) \le \frac{\varepsilon}{3}$. With our choice of $T$, $\ln\left(\frac{1}{\varepsilon}\right) = O(T)$.

If we condition on the event $A$ of the $T_i$'s being a particular sequence $T_1, \ldots, T_n$ with $n < \frac{eT\lambda}{C} + \ln\left(\frac{1}{\varepsilon}\right)$, Pinsker's inequality and Lemma F.1 (with $K = \kappa = \frac{1}{\sigma^2}$) gives us

$$\left\|p^T(\cdot|A) - q^T(\cdot|A)\right\|_1 \le \sqrt{2\mathrm{KL}(p^t(\cdot|A)\|q^t(\cdot|A))} \tag{130}$$

$$= O\left(\max\left\{\frac{\eta^2 D^6 T^2\lambda}{\sigma^{14}\left(\frac{D^2}{\sigma^2} + d\right)}, \eta^2 D^3\frac{1}{\sigma^6}D^2, \eta D^2\frac{1}{\sigma^4}dT\right\}\right) \tag{131}$$

In order for this to be $\le \frac{\varepsilon}{3}$, we need (for some absolute constant $C_1$)

$$\eta \le \frac{C_1\sigma^3\varepsilon}{D^2}\min\left\{\frac{\sigma^4}{\left(\frac{D}{\sigma} + \sqrt{d}\right)T}, \frac{1}{D^{\frac{1}{2}}}, \frac{\sigma\varepsilon}{dT}\right\}. \tag{132}$$

Putting everything together,

$$\left\|p^T - q^T\right\|_1 \le \mathbb{P}(N_T \ge cT\lambda) + \left\|p^t(\cdot|N_T \ge cT\lambda) - q^t(\cdot|N_T \ge cT\lambda)\right\|_1 \le \frac{\varepsilon}{3} + \frac{\varepsilon}{3} = \frac{2\varepsilon}{3}. \tag{133}$$

This gives $\left\|p - q^T\right\|_1 \le \varepsilon$.

For the second part, setting $\varepsilon = O\left(\frac{1}{\ell L}\right)$ gives that $\|p_\ell - q_\ell^T\| = O\left(\frac{1}{L}\right)$. We will apply Lemma G.1. By Lemma L.14 the assumptions of Lemma G.1 are satisfied with $C = O(1)$, as we have

$$\frac{\beta_{i+1} - \beta_i}{\beta_i} = O\left(\frac{1}{\alpha\frac{D^2}{\sigma^2} + d + \ln\left(\frac{1}{w_{\min}}\right)}\right). \tag{134}$$

By Lemma G.1, after collecting $n = \Omega\left(L^2 \ln\left(\frac{1}{\delta}\right)\right)$ samples, with probability $\geq 1 - \delta$, $\left|\frac{\widehat{Z_{\ell+1}/Z_\ell}}{Z_{\ell+1}/Z_\ell} - 1\right| \leq \frac{1}{L}$. Set $\widehat{Z_{\ell+1}} = \bar{r}\widehat{Z_\ell}$. Then $\frac{\widehat{Z_{\ell+1}}}{\widehat{Z_\ell}} \in [1 - \frac{1}{L}, 1 + \frac{1}{L}]\frac{Z_{\ell+1}}{Z_\ell}$ and $\frac{\widehat{Z_{\ell+1}}}{\widehat{Z_1}} \in \left[\left(1 - \frac{1}{L}\right)^\ell, \left(1 + \frac{1}{L}\right)^\ell\right]\frac{Z_{\ell+1}}{Z_1}$. $\qquad\square$

*Proof of Theorem A.2.* Choose $\delta = \frac{\varepsilon}{2L}$ where $L$ is the number of temperatures. Use Lemma G.2 inductively, with probability $1 - \frac{\varepsilon}{2}$ each estimate satisfies $\frac{\widehat{Z_\ell}}{\widehat{Z_1}} \in [\frac{1}{e}, e]$. Estimating the final distribution within $\frac{\varepsilon}{2}$ accuracy gives the desired sample. $\qquad\square$

# H  General log-concave densities

In this section we generalize the main theorem from gaussian to log-concave densities.

## H.1  Simulated tempering for log-concave densities

First we rework Section E for log-concave densities.

**Theorem H.1** (cf. Theorem E.1)**.** *Suppose $f_0$ satisfies Assumption A.1(2) ($f_0$ is $\kappa$-strongly convex, $K$-smooth, and has minimum at 0).*

*Let $M$ be the continuous simulated tempering chain for the distributions*

$$p_i \propto \sum_{j=1}^{m} w_j e^{-\beta_i f_0(x - \mu_j)} \tag{135}$$

*with rate $\Omega\left(\frac{r}{D^2}\right)$, relative probabilities $r_i$, and temperatures $0 < \beta_1 < \cdots < \beta_L = 1$ where*

$$D = \max\left\{\max_j \|\mu_j\|, \frac{\kappa^{\frac{1}{2}}}{d^{\frac{1}{2}}K}\right\} \tag{136}$$

$$\beta_1 = \Theta\left(\frac{\kappa}{dK^2 D^2}\right) \tag{137}$$

$$\frac{\beta_{i+1}}{\beta_i} \leq 1 + \frac{\kappa}{Kd\left(\ln\left(\frac{K}{\kappa}\right) + 1\right)} \tag{138}$$

$$L = \Theta\left(\frac{Kd\left(\ln\left(\frac{K}{\kappa}\right) + 1\right)^2}{\kappa} \ln\left(\frac{dKD}{\kappa}\right)\right) \tag{139}$$

$$r = \frac{\min_i r_i}{\max_i r_i}. \tag{140}$$

*Then $M$ satisfies the Poincaré inequality*

$$\mathrm{Var}(g) \leq O\left(\frac{L^2 D^2}{r^2}\right)\mathscr{E}(g, g) = O\left(\frac{K^2 D^2 \ln\left(\ln\left(\frac{K}{\kappa}\right) + 1\right)^4 \ln\left(\frac{dKD}{\kappa}\right)^2}{\kappa^2 r^2}\right)\mathscr{E}(g, g). \tag{141}$$

*Proof.* Note that forcing $D \leq \frac{\kappa^{\frac{1}{2}}}{d^{\frac{1}{2}}K}$ ensures $\beta_1 = \Omega(1)$.

The proof follows that of Theorem E.1, except that we need to use Lemmas L.12 and L.11 to bound the $\chi^2$-divergences. Steps 1 and 2 are the same: we consider the decomposition where $p_{i,j} \propto e^{-\beta_i f_0(x - \mu_j)}$ and note $\mathscr{E}_{i,j}$ satisfies the Poincaré inequality

$$\mathrm{Var}_{p_{i,j}}(g_i) \leq \frac{1}{\kappa\beta_i}\mathscr{E}_{i,j} = O(D^2)\mathscr{E}_{i,j}(g_i, g_i). \tag{142}$$

By Lemma L.12,

$$\chi^2(p_{i,j}\|p_{i-1,j}) \leq e^{\frac{1}{2}\left|1 - \frac{\beta_{i-1}}{\beta_i}\right|\frac{Kd}{\kappa - K\left|1 - \frac{\beta_{i-1}}{\beta_i}\right|}\left(\sqrt{\ln\left(\frac{K}{\kappa}\right)} + 5\right)^2} \left(\left(1 - \frac{K}{\kappa}\left|1 - \frac{\beta_{i-1}}{\beta_i}\right|\right)\left(1 + \left|1 - \frac{\beta_{i-1}}{\beta_i}\right|\right)\right)^{-\frac{d}{2}} - 1 \tag{143}$$

$$= O(1). \tag{144}$$

By Lemma L.11,

$$\chi^2(p_{1,j}\|p_{1,j'}) \le e^{\frac{1}{2}\beta_1\kappa(2D)^2+\sqrt{\beta_1}K(2D)\sqrt{\frac{d}{\kappa}}\left(\sqrt{\ln\left(\frac{K}{\kappa}\right)}+5\right)} \tag{145}$$

$$\cdot\left(e^{K(2D)\sqrt{\frac{d}{\kappa}}}+\sqrt{\beta_1}K(2D)\sqrt{\frac{4\pi}{\kappa}}e^{\frac{2\sqrt{\beta_1}K(2D)\sqrt{d}}{\sqrt{\kappa}}+\frac{\beta_1 K^2(2D)^2}{2\kappa}}\right)-1=O(1). \tag{146}$$

The rest of the proof is the same. $\qquad\square$

**Theorem H.2** (cf. Theorem E.6). *Suppose $f_0$ satisfies Assumption A.1(2) ($f_0$ is $\kappa$-strongly convex, $K$-smooth, and has minimum at 0).*

*Suppose $\sum_{j=1}^m w_j = 1$, $w_{\min} = \min_{1\le j\le m} w_i > 0$, and $D = \max_{1\le j\le m}\|\mu_j\|$. Let $M$ be the continuous simulated tempering chain for the distributions*

$$p_i \propto \left(\sum_{j=1}^m w_j e^{-f_0(x-\mu_j)}\right)^{\beta_i} \tag{147}$$

*with rate $O\left(\frac{r}{D^2}\right)$, relative probabilities $r_i$, and temperatures $0 < \beta_1 < \cdots < \beta_L = 1$ satisfying the same conditions as in Theorem H.1. Then $M$ satisfies the Poincaré inequality*

$$\mathrm{Var}(g) \le O\left(\frac{L^2 D^2}{r^2 w_{\min}^2}\right)\mathscr{E}(g,g) = O\left(\frac{K^2 d^2\left(\ln\left(\frac{K}{\kappa}\right)+1\right)^4 \ln\left(\frac{dKD}{\kappa}\right)^2}{\kappa^2 r^2 w_{\min}^2}\right)\mathscr{E}(g,g). \tag{148}$$

*Proof.* Let $\widetilde{p}_i$ be the probability distributions in Theorem E.1 with the same parameters as $p_i$ and let $\widetilde{p}$ be the stationary distribution of that simulated tempering chain. By Theorem H.1, $\mathrm{Var}_{\widetilde{p}}(g) = O\left(\frac{L^2 D^2}{r^2}\right)\mathscr{E}\widetilde{p}(g,g)$. Now use By Lemma E.3, $\frac{p_i}{\widetilde{p}_i} \in \left[1, \frac{1}{w_{\min}}\right]\frac{\widetilde{Z}_i}{Z_i}$. Now use Lemma E.5 with $e^\Delta = \frac{1}{w_{\min}}$. $\qquad\square$

## H.2 Proof of main theorem for log-concave densities

Next we rework Section G for log-concave densities, and prove the main theorem for log-concave densities, Theorem A.3.

**Lemma H.3** (cf. Lemma G.2). *Suppose $f_0$ satisfies Assumption A.1(2) ($f_0$ is $\kappa$-strongly convex, $K$-smooth, and has minimum at 0).*

*Suppose that Algorithm 1 is run on $f(x) = -\ln\left(\sum_{j=1}^m w_j f_0(x-\mu_j)\right)$ with temperatures $0 < \beta_1 < \cdots < \beta_\ell \le 1$, $\ell \le L$ with partition function estimates $\widehat{Z}_1,\ldots,\widehat{Z}_\ell$ satisfying*

$$\left|\frac{\widehat{Z}_i}{Z_i}\Big/\frac{\widehat{Z}_1}{Z_1}\right| \in \left[\left(1-\frac{1}{L}\right)^{i-1}, \left(1+\frac{1}{L}\right)^{i-1}\right] \tag{149}$$

*for all $1 \le i \le \ell$. Suppose $\sum_{j=1}^m w_j = 1$, $w_{\min} = \min_{1\le j\le m} w_i > 0$, and $D = \max\left\{\max_j\|\mu_j\|, \frac{\kappa^{\frac{1}{2}}}{d^{\frac{1}{2}}K}\right\}$, $K \ge 1$, and the parameters satisfy*

$$\lambda = \Theta\left(\frac{1}{D^2}\right) \tag{150}$$

$$\beta_1 = O\left(\frac{\kappa}{dK^2 D^2}\right) \tag{151}$$

$$\frac{\beta_{i+1}}{\beta_i} \le 1 + \frac{\kappa}{Kd\left(\ln\left(\frac{K}{\kappa}\right)+1\right)} \tag{152}$$

$$L = \Theta\left(\frac{Kd\left(\ln\left(\frac{K}{\kappa}\right)+1\right)^2}{\kappa}\ln\left(\frac{dKD}{\kappa}\right)\right) \tag{153}$$

$$T = \left( \frac{L^2 D^2}{w_{\min}^2} d \ln\left( \frac{\ell}{\varepsilon w_{\min}} \right) \ln\left( \frac{K}{\kappa} \right) \right) \tag{154}$$

$$\eta = O\left( \min\left\{ \frac{\varepsilon}{D^2 K^{\frac{7}{2}} \left( D \frac{K^{\frac{1}{2}}}{\kappa^{\frac{1}{2}}} + d^{\frac{1}{2}} \right) T}, \frac{\varepsilon}{D^{\frac{5}{2}} K^{\frac{3}{2}} \left( \left(\frac{K}{\kappa}\right)^{\frac{1}{2}} + 1 \right)}, \frac{\varepsilon}{D^2 K^2 dT} \right\} \right). \tag{155}$$

*Let $q^0$ be the distribution $\left( N\left(0, \frac{1}{\kappa \beta_1}\right), 1 \right)$ on $[\ell] \times \mathbb{R}^d$. The distribution $q^T$ after running time $T$ satisfies $\left\| p - q^T \right\|_1 \le \varepsilon$.*

*Setting $\varepsilon = O\left(\frac{1}{\ell L}\right)$ above and taking $n = \Omega\left(L^2 \ln\left(\frac{1}{\delta}\right)\right)$ samples, with probability $1 - \delta$ the estimate*

$$\widehat{Z}_{\ell+1} = \bar{r}\widehat{Z}_\ell, \qquad\qquad \bar{r} := \left( \frac{1}{n} \sum_{j=1}^{n} e^{(-\beta_{\ell+1} + \beta_\ell) f_i(x_j)} \right) \tag{156}$$

*also satisfies* (162).

*Proof.* Begin as in the proof of Lemma G.2. Let $p_{\beta,i} \propto e^{-\beta_1 f_0(x - \mu_i)}$ be a probability density function.

Write $\left\| p - q^T \right\|_1 \le \left\| p - p^T \right\|_1 + \left\| p^T - q^T \right\|_1$. Bound the first term by $\left\| p - p^T \right\|_1 \le \sqrt{\chi^2(p\|p^T)} \le e^{-\frac{T}{2C}} \sqrt{\chi^2(p\|p^0)}$ where $C$ is the upper bound on the Poincaré constant in Theorem H.2. As in (128), we get

$$\chi^2(p\|p^0) = O\left( \frac{\ell}{w_{\min}} \right) \left( 1 + \sum_{j=1}^{m} w_j \chi^2\left( p_{\beta_1,j} \big\| N\left(0, \frac{1}{\kappa \beta_1} I_d\right) \right) \right). \tag{157}$$

By Lemma L.13 with strong convexity constants $\kappa \beta_1$ and $K \beta_1$, this is

$$O\left( \frac{\ell}{w_{\min}} \left( \frac{K}{\kappa} \right)^{\frac{d}{2}} e^{K \beta_1 D^2} \right) = O\left( \frac{\ell}{w_{\min}} \left( \frac{K}{\kappa} \right)^{\frac{d}{2}} \right) \tag{158}$$

when $\beta_1 = O\left(\frac{K}{D^2}\right)$. Thus for $T = \Omega\left( C \ln\left(\frac{\ell}{\varepsilon w_{\min}}\right) d \ln\left(\frac{K}{\kappa}\right) \right)$, $\left\| p - p^T \right\|_1 \le \frac{\varepsilon}{3}$.

Again conditioning on the event $A$ that $N_T = \max\{n : T_n \le T\} = O(T\lambda)$, we get by Lemma F.1 that

$$\left\| p^T(\cdot|A) - q^T(\cdot|A) \right\|_1 = O\left( \eta^2 D^6 K^7 \left( D^2 \frac{K^2}{\kappa} + d \right) Tn + \eta^2 D^5 \left( \frac{K}{\kappa} + 1 \right) + \eta D^2 K^2 dT \right). \tag{159}$$

Choosing $\eta$ as in the problem statement, we get $\left\| p - q^T \right\|_1 \le \varepsilon$ as before. Finally, apply Lemma G.1, checking the assumptions are satisfied using Lemma L.15. The assumptions of Lemma L.15 hold, as

$$\frac{\beta_{i+1} - \beta_i}{\beta_i} = O\left( \frac{1}{\alpha K D^2 + \frac{d}{\kappa}\left(1 + \ln\left(\frac{K}{\kappa}\right)\right) + \frac{1}{\kappa} \ln\left(\frac{1}{w_{\min}}\right)} \right). \tag{160}$$

$\square$

*Proof of Theorem A.3.* This follows from Lemma H.3 in exactly the same way that the main theorem for gaussians (Theorem A.2) follows from Lemma G.2. $\square$

# I Perturbation tolerance

The proof of Theorem A.4 will follow immediately from Lemma I.2, which is a straightforward analogue of Lemma G.2.

## I.1 Simulated tempering for distribution with perturbation

First, we consider the mixing time of the continuous tempering chain, analogously to Theorem H.2:

**Theorem I.1** (cf. Theorem H.2). *Suppose $f_0$ satisfies Assumption A.1*

*Let $M$ be the continuous simulated tempering chain with rate $O\left(\frac{r}{D^2}\right)$, relative probabilities $r_i$, and temperatures $0 < \beta_1 < \cdots < \beta_L = 1$ satisfying the same conditions as in Lemma I.2. Then $M$ satisfies the Poincaré inequality*

$$\mathrm{Var}(g) \le O\left(\frac{L^2 D^2}{r^2 w_{\min}^2}\right) \mathscr{E}(g,g) = O\left(\frac{K^2 d^2 \left(\ln\left(\frac{K}{\kappa}\right)+1\right)^4 \ln\left(\frac{dKD}{\kappa}\right)^2}{\kappa^2 r^2 e^{2\Delta} w_{\min}^2}\right) \mathscr{E}(g,g). \quad (161)$$

*Proof.* The proof is almost the same as Let $\widetilde{p}_i$ be the probability distributions in Theorem E.1 with the same parameters as $p_i$ and let $\widetilde{p}$ be the stationary distribution of that simulated tempering chain. By Theorem H.1, $\mathrm{Var}_{\widetilde{p}}(g) = O\left(\frac{L^2 D^2}{r^2}\right) \mathscr{E}\widetilde{p}(g,g)$. Now use By Lemma E.3, $\frac{p_i}{\widetilde{p}_i} \in \left[1, \frac{1}{w_{\min}}\right] \frac{\widetilde{Z}_i}{Z_i}$. Now use Lemma E.5 with $e^{\Delta}$ substituted to be $e^{\Delta} \frac{1}{w_{\min}}$. $\qquad\square$

## I.2 Proof of main theorem with perturbations

**Lemma I.2** (cf. Lemma H.3). *Suppose that Algorithm 1 is run on $f(x) = -\ln\left(\sum_{j=1}^m w_j f_0(x - \mu_j)\right)$ with temperatures $0 < \beta_1 < \cdots < \beta_\ell \le 1$, $\ell \le L$ with partition function estimates $\widehat{Z}_1, \ldots, \widehat{Z}_\ell$ satisfying*

$$\left|\frac{\widehat{Z}_i}{Z_i} \Big/ \frac{\widehat{Z}_1}{Z_1}\right| \in \left[\left(1 - \frac{1}{L}\right)^{i-1}, \left(1 + \frac{1}{L}\right)^{i-1}\right] \quad (162)$$

*for all $1 \le i \le \ell$. Suppose $\sum_{j=1}^m w_j = 1$, $w_{\min} = \min_{1 \le j \le m} w_i > 0$, and $D = \max\left\{\max_j \|\mu_j\|, \frac{\kappa^{\frac{1}{2}}}{d^{\frac{1}{2}} K}\right\}$, $K \ge 1$, and the parameters satisfy*

$$\lambda = \Theta\left(\frac{1}{D^2}\right) \quad (163)$$

$$\beta_1 = O\left(\min\left\{\Delta, \frac{\kappa}{dK^2 D^2}\right\}\right) \quad (164)$$

$$\frac{\beta_{i+1}}{\beta_i} \le \min\left\{\Delta, 1 + \frac{\kappa}{Kd\left(\ln\left(\frac{K}{\kappa}\right)+1\right)}\right\} \quad (165)$$

$$L = \Theta\left(\frac{Kd\left(\ln\left(\frac{K}{\kappa}\right)+1\right)^2}{\kappa} \ln\left(\frac{dKD}{\kappa}\right)\right) \quad (166)$$

$$T = \left(e^{2\Delta} \frac{L^2 D^2}{w_{\min}^2} d\ln\left(\frac{\ell}{\varepsilon w_{\min}}\right) \ln\left(\frac{K}{\kappa}\right)\right) \quad (167)$$

$$\eta = O\left(\min\left\{\frac{\varepsilon}{D^2(K+\tau)^{\frac{7}{2}}\left(D\frac{K+\tau}{\kappa^{\frac{1}{2}}}+d^{\frac{1}{2}}\right)T}, \frac{\varepsilon}{D^{\frac{5}{2}}(K+\tau)^{\frac{3}{2}}\left(\left(\frac{K+\tau}{\kappa}\right)^{\frac{1}{2}}+1\right)}, \frac{\varepsilon}{D^2(K+\tau)^2 dT}\right\}\right). \quad (168)$$

*Let $q^0$ be the distribution $\left(N\left(0, \frac{1}{\kappa\beta_1}\right), 1\right)$ on $[\ell] \times \mathbb{R}^d$. The distribution $q^T$ after running time $T$ satisfies $\left\|p - q^T\right\|_1 \le \varepsilon$.*

*Setting $\varepsilon = O\left(\frac{1}{\ell L}\right)$ above and taking $n = \Omega\left(L^2 \ln\left(\frac{1}{\delta}\right)\right)$ samples, with probability $1 - \delta$ the estimate*

$$\widehat{Z}_{\ell+1} = \bar{r}\widehat{Z}_\ell, \qquad \bar{r} := \left(\frac{1}{n}\sum_{j=1}^n e^{(-\beta_{\ell+1}+\beta_\ell)f_i(x_j)}\right) \quad (169)$$

*also satisfies* (162).

The way we prove this theorem is to prove the tolerance of each of the proof ingredients to perturbations to $f$.

### I.2.1 Discretization

We now verify all the discretization lemmas continue to hold with perturbations.

The proof of Lemma F.3, combined with the fact that $\left\|\nabla\tilde{f} - \nabla f\right\|_\infty \leq \Delta$ gives

**Lemma I.3** (Perturbed reach of continuous chain)**.** *Let $P_T^\beta(X)$ be the Markov kernel corresponding to evolving Langevin diffusion*

$$\frac{dX_t}{dt} = -\beta\nabla\tilde{f}(X_t) + dB_t$$

*with $f$ and $D$ are as defined in 2 for time $T$. Then,*

$$\mathbb{E}[\|X_t - x^*\|^2] \lesssim \mathbb{E}[\|X_0 - x^*\|^2] + \left(400\beta\frac{D^2K^2\tau^2}{\kappa} + d\right)T$$

*Proof.* The proof proceeds exactly the same as Lemma F.3. □

Furthermore, since $\|\nabla^2\tilde{f}(x) - \nabla^2 f(x)\|_2 \leq \tau, \forall x \in \mathbb{R}^d$, from Lemma F.4, we get

**Lemma I.4** (Perturbed Hessian bound)**.**

$$\|\nabla^2\tilde{f}(x)\|_2 \leq 4(DK)^2 + \tau, \forall x \in \mathbb{R}^d$$

As a consequence, the analogue of Lemma F.5 gives:

**Lemma I.5** (Bounding interval drift)**.** *In the setting of Lemma F.5, let $x \in \mathbb{R}^d, i \in [L]$, and let $\eta \leq \frac{(\frac{1}{\sigma}+\tau)^2}{\alpha}$. Then,*

$$KL(P_T(x,i)\|\widehat{P_T}(x,i)) \leq \frac{4D^6\eta^7(K+\tau)^7}{3}\left(\|x - x^*\|_2^2 + 8Td\right) + dTD^2\eta(K+\tau)^2$$

Putting these together, we get the analogue of Lemma F.1:

**Lemma I.6.** *Fix times $0 < T_1 < \cdots < T_n \leq T$.*

*Let $p^T, q^T : [L] \times \mathbb{R}^d \to \mathbb{R}$ be defined as follows.*

1. *$p^T$ is the continuous simulated tempering Markov as in Definition C.1 but with fixed transition times $T_1, \ldots, T_n$. The component chains are Langevin diffusions on $p_i \propto \left(\sum_{j=1}^m w_j e^{-f_0(x-\mu_i)}\right)^{\beta_i}$.*

2. *$q^T$ is the discretized version as in Algorithm (1), again with fixed transition times $T_1, \ldots, T_n$, and with step size $\eta \leq \frac{\sigma^2}{2}$.*

*Then*

$$KL(p^T\|q^T) \lesssim \eta^2 D^6(K+\tau)^6\left(D^2\frac{(K+\tau)^2}{\kappa} + d\right)Tn + \eta^2 D^3(K+\tau)^3\max_i\mathbb{E}_{x\sim p^0(\cdot,i)}\|x-x^*\|_2^2 + \eta D^2(K+\tau)^2dT$$

*where $x^*$ is the maximum of $\sum_{j=1}^m w_j e^{-f_0(x-\mu_j)}$ and satisfies $\|x^*\| = O(D)$ where $D = \max\|\mu_j\|$.*

### I.2.2 Putting it all together

Finally, we prove Lemma I.2.

*Proof of Lemma I.2.* The proof is analogous to the one of Lemma G.2 in combination with the Lemmas from the previous subsections, so we just point out the differences.

We bound $\chi^2(\tilde{p}||q^0)$ as follows: by the proof of Lemma G.2, we have $\chi^2(p||q^0) = O\left(\frac{\ell}{w_{\min}}K^{\frac{d}{2}}\right)$. By the definition of $\chi^2$, this means

$$\int \frac{q^0(x)^2}{p(x)}dx \leq O\left(\frac{\ell}{w_{\min}}K^{\frac{d}{2}}\right)$$

This in turn implies that

$$\chi^2(\tilde{p}||q^0) \leq \int \frac{(q^0(x))^2}{\tilde{p}(x)}dx \leq O\left(\frac{\ell}{w_{\min}}K^{\frac{d}{2}}e^{\Delta}\right)$$

Then, analogously as in Lemma G.2, we get

$$\left\|p^T(\cdot|A) - q^T(\cdot|A)\right\|_1 = O\left(\eta^2 D^6 (K+\tau)^7 \left(D^2\frac{(K+\tau)^2}{\kappa} + d\right)T\eta + \eta^2 D^5\left(\frac{K}{\kappa}+1\right) + \eta D^2(K+\tau)^2 dT\right).$$
(170)

Choosing $\eta$ as in the statement of the lemma, $\left\|p - q^T\right\|_1 \leq \varepsilon$ follows. The rest of the lemma is identical to Lemma G.2.

$\square$

# J   Examples

It might be surprising that sampling a mixture of gaussians require a complicated Markov Chain such as simulated tempering. However, many simple strategies seem to fail.

**Langevin with few restarts**   One natural strategy to try is simply to run Langevin a polynomial number of times from randomly chosen locations. While the time to "escape" a mode and enter a different one could be exponential, we may hope that each of the different runs "explores" the individual modes, and we somehow stitch the runs together. The difficulty with this is that when the means of the gaussians are not well-separated, it's difficult to quantify how far each of the individual runs will reach and thus how to combine the various runs.

**Recovering the means of the gaussians**   Another natural strategy would be to try to recover the means of the gaussians in the mixture by performing gradient descent on the log-pdf with a polynomial number of random restarts. The hope would be that maybe the local minima of the log-pdf correspond to the means of the gaussians, and with enough restarts, we should be able to find them.

Unfortunately, this strategy without substantial modifications also seems to not work: for instance, in dimension $d$, consider a mixture of $d+1$ gaussians, $d$ of them with means on the corners of a $d$-dimensional simplex with a side-length substantially smaller than the diameter $D$ we are considering, and one in the center of the simplex. In order to discover the mean of the gaussian in the center, we would have to have a starting point extremely close to the center of the simplex, which in high dimensions seems difficult.

Additionally, this doesn't address at all the issue of robustness to perturbations. Though there are algorithms to optimize "approximately" convex functions, they can typically handle only very small perturbations. [BLNR15, LR16]

**Gaussians with different covariance**   Our result requires all the gaussians to have the same variance. This is necessary, as even if the variance of the gaussians only differ by a factor of 2, there are examples where a simulated tempering chain takes exponential time to converge [WSH+09]. Intuitively, this is illustrated in Figure 1. The figure on the left shows the distribution in low temperature – in this case the two modes are separate, and both have a significant mass. The figure on the right shows the distribution in high temperature. Note that although in this case the two modes are connected, the volume of the mode with smaller variance is much smaller (exponentially small in

$d$). Therefore in high dimensions, even though the modes can be connected at high temperature, the probability mass associated with a small variance mode is too small to allow fast mixing.

In the next section, we show that even if we do not restrict to the particular simulated tempering chain, no efficient algorithm can efficiently and robustly sample from a mixture of two Gaussians with different covariances.

Figure 1: Mixture of two gaussians with different covariance at different temperature

## K    Lower bound when Gaussians have different variance

In this section, we give a lower bound showing that in high dimensions, if the Gaussians can have different covariance matrices, results similar to our Theorem A.2 cannot hold. In particular, we construct a log density function $\tilde{f}$ that is close to the log density of mixture of two Gaussians (with different variances), and show that any algorithm must query the function at exponentially many locations in order to sample from the distribution. More precisely, we prove the following theorem:

**Theorem K.1.** *There exists a function $\tilde{f}$ such that $\tilde{f}$ is close to a negative log density function $f$ for a mixture of two Gaussians: $\left\| f - \tilde{f} \right\|_{\infty} \leq \log 2$, $\forall x \, \|\nabla f(x) - \nabla \tilde{f}(x)\| \leq O(d)$, $\|\nabla^2 f(x) - \nabla^2 \tilde{f}(x)\| \leq O(d)$. Let $\tilde{p}$ be the distribution whose density function is proportional to $\exp(-\tilde{f})$. There exists constant $c > 0, C > 0$, such that when $d \geq C$, any algorithm with at most $2^{cd}$ queries to $\tilde{f}$ and $\nabla \tilde{f}$ cannot generate a distribution that is within TV-distance $0.3$ to $\tilde{p}$.*

In order to prove this theorem, we will first specify the mixture of two Gaussians. Consider a uniform mixture of two Gaussian distributions $N(0, 2I)$ and $N(u, I)(u \in \mathbb{R}^d)$ in $\mathbb{R}^d$.

**Definition K.2.** *Let $f_1 = \|x\|^2/4 + \frac{d}{2}\log(2\sqrt{2}\pi)$ and $f_2 = \|x - u\|^2/2 + \frac{d}{2}\log(2\pi)$. The mixture $f$ used in the lower bound is*

$$f = -\log(\frac{1}{2}(e^{-f_1} + e^{-f_1})).$$

In order to prove the lower bound, we will show that there is a function $\tilde{f}$ close to $f$, such that $\tilde{f}$ behaves exactly like a single Gaussian $N(0, 2I)$ on almost all points. Intuitively, any algorithm with only queries to $\tilde{f}$ will not be able to distinguish it with a single Gaussian, and therefore will not be able to find the second component $N(u, I)$. More precisely, we have

**Lemma K.3.** *When $\|u\| \geq 4d \log 2$, for any point $x$ outside of the ball with center $2u$ and radius $1.5\|u\|$, we have $e^{-f_1(x)} \geq e^{-f_2(x)}$.*

*Proof.* The Lemma follows from simple calculation. In order for $e^{-f_1(x)} \geq e^{-f_2(x)}$, since $e^x$ is monotone we know

$$-\frac{\|x - u\|^2}{2} \leq -\frac{\|x\|^2}{4} - \frac{d}{4}\log 2.$$

This is a quadratic inequality in terms of $x$, reordering the terms we get

$$\|x - 2u\|^2 \geq d \log 2 + 2\|u\|^2.$$

Since $d \log 2 \leq 0.25\|u\|^2$, we know whenever $\|x - 2u\|^2 \geq 1.5\|u\|$ this is always satisfied, and hence $e^{-f_1(x)} \geq e^{-f_2(x)}$. $\qquad \square$

The lemma shows that outside of this ball, the contribution from the first Gaussian is dominating. Intuitively, we try to make $\tilde{f} = f_1$ outside of this ball, and $\tilde{f} = f$ inside the ball. To make the function continuous, we shift between the two functions gradually. More precisely, we define $\tilde{f}$ as follows:

**Definition K.4.** *The function*

$$\tilde{f}(x) = g(x)f_1(x) + (1 - g(x))f(x). \tag{171}$$

*Here the function $g(x)$ (see Definition K.6) satisfies*

$$g(x) = \begin{cases} 1 & \|x - 2u\| \geq 1.6\|u\| \\ 0 & \|x - 2u\| \leq 1.5\|u\| \\ \in [0,1] & \text{otherwise} \end{cases}$$

*Also $g(x)$ is twice differentiable with all first and second order derivatives bounded.*

With a carefully constructed $g(x)$, it is possible to prove that $\tilde{f}$ is point-wise close to $f$ in function value, gradient and Hessian, as stated in the Lemma below. Since these are just routine calculations, we leave the construction of $g(x)$ and verification of this lemma at the end of this section.

**Lemma K.5.** *For the functions $f$ and $\tilde{f}$ defined in Definitions K.2 and K.4, if $\|u\| \geq 4d \log 2$, there exists a large enough constant $C$ such that*

$$|f - \tilde{f}|_\infty \leq \log 2$$
$$\forall x \quad \|\nabla f(x) - \nabla \tilde{f}(x)\| \leq C\|u\|$$
$$\forall x \quad \|\nabla^2 f(x) - \nabla^2 \tilde{f}(x)\| \leq C\|u\|^2.$$

Now we are ready to prove the main theorem:

*Proof of Theorem K.1.* We will show that if we pick $\|u\|$ to be a uniform random vector with norm $8d \log 2$, there exists constant $c > 0$ such that for any algorithm, with probability at least $1 - \exp(-cd)$, in the first $\exp cd$ iterations of the algorithm there will be no vector $x \neq 0$ such that $\cos \theta(x, u) \geq 3/5$.

First, by standard concentration inequalities, we know for any fixed vector $x \neq 0$ and a uniformly random $u$,

$$\Pr[\cos \theta(x, u) \geq 3/5] \leq \exp -c'd,$$

for some constant $c' > 0$ ($c' = 0.01$ suffices).

Now, for any algorithm, consider running the algorithm with oracle to $f_1$ and $\tilde{f}$ respectively (if the algorithm is randomized, we also couple the random choices of the algorithm in these two runs). Suppose when the oracle is $f_1$ the queries are $x_1, x_2, ..., x_t$ and when the oracle is $\tilde{f}$ the queries are $\tilde{x}_1, ..., \tilde{x}_t$.

Let $c = c'/2$, when $t \leq \exp(cd)$, by union bound we know with probability at least $1 - \exp(-cd)$, we have $\cos \theta(x_i, u) < 3/5$ for all $i \leq t$. On the other hand, every point $y$ in the ball with center $2\|u\|$ and radius $1.6\|u\|$ has $\cos \theta(y, u) \geq 3/5$. We know $\|x_i - 2u\| > 1.6\|u\|$, hence $f_1(x_i) = \tilde{f}(x_i)$ for all $i \leq t$ (the derivatives are also the same). Therefore, the algorithm is going to get the same response no matter whether it has access to $f_1$ or $\tilde{f}$. This implies $\tilde{x}_i = x_i$ for all $i \leq t$.

Now, to see why this implies the output distribution of the last point is far from $\tilde{p}$, note that when $d$ is large enough $\tilde{p}$ has mass at least $0.4$ in ball $\|x_i - 2u\| \leq 1.6\|u\|$ (because essentially all the mass in the second Gaussian is inside this ball), while the algorithm has less than $0.1$ probability of having any point in this region. Therefore the TV distance is at least $0.3$ and this finishes the proof. $\qquad \square$

### K.1 Construction of $g$ and closeness of two functions

Now we finish the details of the proof by construction a function $g$.

**Definition K.6.** *Let $h(x)$ be the following function:*

$$h(x) = \begin{cases} 1 & x \geq 1 \\ 0 & x \leq 0 \\ x^2(1-x)^2 + (1 - (1-x)^2)^2 & x \in [0,1] \end{cases}$$

*We then define $g(x)$ to be $g(x) := h\left(10\left(\frac{\|x-2u\|}{\|u\|} - 1.5\right)\right)$.*

For this function we can prove:

**Lemma K.7.** *The function $g$ defined above satisfies*

$$g(x) = \begin{cases} 1 & \|x-2u\| \geq 1.6\|u\| \\ 0 & \|x-2u\| \leq 1.5\|u\| \\ \in [0,1] & otherwise \end{cases}$$

*Also $g(x)$ is twice differentiable. There exists large enough constant $C_g > 0$ such that for all $x$*

$$\|\nabla g(x)\| \leq C_g\|u\| \quad \|\nabla^2 g(x)\| \leq C_g(\|u\|^2 + 1).$$

*Proof.* First we prove properties of $h(x)$. Let $h_0(x) = x^2(1-x)^2 + (1 - (1-x)^2)^2$, it is easy to check that $h_0(0) = h_0'(0) = h_0''(0) = 0$, $h_0(1) = 1$ and $h_0'(1) = h_0''(1) = 0$. Therefore the entire function $h(x)$ is twice differentiable.

Also, we know $h_0'(x) = 2x(4x^2 - 9x + 5)$, which is always positive when $x \in [0,1]$. Therefore $h(x)$ is monotone in $[0,1]$. The second derivative $h_0''(x) = 24x^2 - 36x + 10$. Just using the naive bound (sum of absolute values of individual terms) we can get for any $x \in [0,1]$ $|h_0'(x)| \leq 36$ and $|h''(x)| \leq 60$. (We can of course compute better bounds but it is not important for this proof.)

Now consider the function $g$. We know when $\|x-2u\| \in [1.5, 1.6]\|u\|$,

$$\nabla g(x) = h'\left(10\left(\frac{\|x-2u\|}{\|u\|} - 1.5\right)\right) \cdot 10(x - 2u).$$

Therefore $\|\nabla g(x)\| \leq 36 \times 10 \times \|x-2u\| \leq C_g\|u\|$ (when $C_g$ is a large enough constant).

For the second order derivative, we know

$$\nabla^2 g(x) = 100h''\left(10\left(\frac{\|x-2u\|}{\|u\|} - 1.5\right)\right)(x-2u)(x-2u)^\top + 10h'\left(10\left(\frac{\|x-2u\|}{\|u\|} - 1.5\right)\right)I.$$

Again by bounds on $h'$ and $h''$ we know there exists large enough constants so that $\|\nabla^2 g(x)\| \leq C_g(\|u\|^2 + 1)$.

$\square$

Finally we can prove Lemma K.5.

*Proof of Lemma K.5.* We first show that the function values are close. When $\|x - 2u\| \leq 1.5\|u\|$, by definition $\tilde{f}(x) = f(x)$. When $\|x - 2u\| \geq 1.5\|u\|$, by property of $g$ we know $\tilde{f}(x)$ is between $f(x)$ and $f_1(x)$. Now by Lemma K.3, in this range $e^{-f_1(x)} \geq e^{-f_2(x)}$, so $f_1(x) - \log 2 \leq f(x) \leq f_1(x)$. As a result we know $|\tilde{f}(x) - f(x)| \leq \log 2$.

Next we consider the gradient. Again when $\|x - 2u\| \leq 1.5\|u\|$ the two functions (and all their derivatives) are the same. When $\|x - 2u\| \in [1.5, 1.6]\|u\|$, we have

$$\nabla \tilde{f}(x) = g(x)\nabla f_1(x) + (1 - g(x))\nabla f(x) + (f_1(x) - f(x))\nabla g(x).$$

By Lemma K.7 we have upperbounds for $g(x)$ and $\|\nabla g(x)\|$, also both $\|\nabla f_1(x)\|$, $\|\nabla f(x)\|$ can be easily bounded by $O(1)\|u\|$, therefore $\|\nabla \tilde{f}(x) - \nabla f(x)\| \leq C\|u\|$ for large enough constant $C$.

When $\|x - 2u\| \geq 1.6\|u\|$, we know $\nabla \tilde{f}(x) = \nabla f_1(x)$. Calculation shows

$$\nabla f_1(x) - \nabla f(x) = \frac{e^{-f_2(x)}}{e^{-f_1(x)} + e^{-f_2(x)}}(\nabla f_1(x) - \nabla f_2(x)).$$

When $\|x\| \leq 50\|u\|$, we have $\|\nabla f_1(x) - \nabla f_2(x)\| \leq 2\|x\| + 2\|u\| \leq O(1)\|u\|$. When $\|x\| \geq 50\|u\|$, it is easy to check that $\frac{e^{-f_2(x)}}{e^{-f_1(x)} + e^{-f_2(x)}} \leq \exp -\|x\|^2/5$ and $\|\nabla f_1(x) - \nabla f_2(x)\| \leq 2\|x\|$, therefore in this case the difference in gradient bounded by $\exp(-t^2/5)2t$ which is always small.

Finally we can check the Hessian. Once again when $\|x - 2u\| \leq 1.5\|u\|$ the two functions are the same. When $\|x - 2u\| \in [1.5, 1.6]\|u\|$, we have

$$\begin{aligned}\nabla^2 \tilde{f}(x) =&g(x)\nabla^2 f_1(x) + (1 - g(x))\nabla^2 f(x) \\ &+ (\nabla f_1(x) - \nabla f(x))(\nabla g(x))^\top + (\nabla g(x))(\nabla f_1(x) - \nabla f(x))^\top \\ &+ (f_1 - f)\nabla^2 g(x).\end{aligned}$$

In this case we get bounds for $g(x), \nabla g(x), \nabla^2 g(x)$ from Lemma K.7, $\|\nabla f_1(x)\|, \|\nabla f(x)\|$ can still be bounded by $O(1)\|u\|$, $\|\nabla^2 f(x)\|, \|\nabla^2 f_1(x)\|$ can be bounded by $O(\|u\|^2)$ and $O(1)$ respectively. Therefore we know $\|\nabla^2 \tilde{f}(x) - \nabla^2 f(x)\| \leq C\|u\|^2$ for large enough constant $C$.

When $\|x - 2u\| \geq 1.6\|u\|$, we have $\tilde{f}(x) = f_1(x)$, and

$$\begin{aligned}\nabla^2 f_1(x) - \nabla^2 f(x) =&\frac{e^{-f_2(x)}}{e^{-f_1(x)} + e^{-f_2(x)}}(\nabla^2 f_1(x) - \nabla^2 f_2(x)) \\ &+ \frac{e^{-f_1(x)-f_2(x)}(\nabla f_1(x) - \nabla f_2(x))(\nabla f_1(x) - \nabla f_2(x))^\top}{(e^{-f_1(x)} + e^{-f_2(x)})^2}.\end{aligned}$$

Here the first term is always bounded by a constant (because $e^{-f_2(x)}$ is smaller, and $\nabla^2 f_1(x) - \nabla^2 f_2(x) = I/2$). For the second term, by arguments similar as before, we know when $\|x\| \leq 50\|u\|$ this is bounded by $O(1)\|u\|^2$. When $\|x\| \geq 50\|u\|$ we can check $\frac{e^{-f_1(x)-f_2(x)}}{(e^{-f_1(x)} + e^{-f_2(x)})^2} \leq \exp(-\|x\|^2/5)$ and $\|(\nabla f_1(x) - \nabla f_2(x))(\nabla f_1(x) - \nabla f_2(x))^\top\| \leq 4\|x\|^2$. Therefore the second term is bounded by $\exp(-t^2/5) \cdot 4t^2$ which is no larger than a constant. Combining all the cases we know there exists a large enough constant $C$ such that $\|\nabla^2 f(x) - \nabla^2 f(x)\| \leq C\|u\|^2$ for all $x$.

$\square$

# L   Calculations on probability distributions

## L.1   Inequalities

**Lemma L.1.** *Let $p, q$ be probability distributions on $\Omega$ and $g : \Omega \to \mathbb{R}$. Then*

$$\left(\int_\Omega gp - \int_\Omega gq\right)^2 \leq \mathrm{Var}_p(g)\chi^2(p\|q). \tag{172}$$

*Proof.*

$$\left(\int_\Omega gp - \int_\Omega gq\right)^2 = \left(\int_\Omega (g - \mathbb{E}_p g)(p - q)\right)^2 \tag{173}$$

$$\leq \left(\int_\Omega (g - \mathbb{E}_p g)^2 p\right)\left(\int_\Omega \frac{(p - q)^2}{p}\right) \tag{174}$$

$$= \mathrm{Var}_p(g)\chi^2(p\|q). \tag{175}$$

$\square$

**Remark L.2.** *Note that this lemma still works when $p, q$ aren't normalized. The only difference is that we replace $\mathbb{E}_p$ by $\mathbb{E}_{\tilde{p}}$ where $\tilde{p} = \frac{p}{\int p}$, in defining $\mathrm{Var}_p(g) = \int \left(g - \mathbb{E}_{\tilde{p}} g\right)^2 p$. (Note we don't normalize the integral.)*

**Lemma L.3.** *Let $p_1, p_2$ be probability distributions on $\Omega$, $g_1, g_2 : \Omega \to \mathbb{R}$, and $q = \min\{p_1, p_2\}$. Then*

$$\left( \int_\Omega g_1 p_1 - \int_\Omega g_2 p_2 \right)^2 \le 6 \operatorname{Var}_{p_1}(g_1) \chi^2(p_1 \| p_2) + 3 \int_\Omega (g_1 - g_2)^2 q \int q. \tag{176}$$

*Proof.* By Lemma L.1 (and Remark L.1) we have

$$\left( \int_\Omega g_1 p_1 - \int_\Omega g_2 p_2 \right)^2 \le \left( \int_\Omega g_1 (p_1 - q) + \int_\Omega (g_1 - g_2) q + \int_\Omega g_2 (q - p_2) \right)^2 \tag{177}$$

$$\le 3 \left[ \left( \int_\Omega g_1 (p_1 - q) \right)^2 + \left( \int_\Omega g_2 (q - p_2) \right)^2 + \int_\Omega (g_1 - g_2) q \right] \tag{178}$$

$$\le 3 \left[ \operatorname{Var}_{p_1}(g_1) \chi^2(p_1 \| q) + \operatorname{Var}_q(g_2) \chi^2(q \| p_2) + \int (g_1 - g_2)^2 q \int q \right] \tag{179}$$

$$\le 6 \operatorname{Var}_{p_1}(g_1) \chi^2(p_1 \| p_2) + 3 \int (g_1 - g_2)^2 q \int q \tag{180}$$

where in the last step we note that

$$\operatorname{Var}_q(g_2) \le \operatorname{Var}_{p_1}(g_2) \tag{181}$$

$$\chi^2(p_1 \| q) = \int_{\Omega, p_2 < p_1} \frac{(p_1 - \min\{p_1, p_2\})^2}{p_1} \le \int \frac{(p_1 - p_2)^2}{p_1} = \chi^2(p_1 \| p_2) \tag{182}$$

$$\chi^2(q \| p_2) = \int_{\Omega, p_1 < p_2} \frac{(\min\{p_1, p_2\} - p_2)^2}{q} \le \int \frac{(p_1 - p_2)^2}{p_1} = \chi^2(p_1 \| p_2). \tag{183}$$

• $\qquad\qquad\qquad\qquad\qquad\qquad\qquad\qquad\qquad\qquad\qquad\qquad\qquad\qquad\qquad\quad$ $\square$

**Lemma L.4.** *If $p = \sum_{i=1}^n w_i p_i$ where $p_i$ are probability distributions and $w_i > 0$ sum to 1, then*

$$\chi^2(p \| q) \le \sum_{i=1}^n w_i \chi^2(p_i \| q). \tag{184}$$

*Proof.* We calculate

$$\chi^2(p \| q) = \sum_{i=1}^n \frac{q(x)^2}{\sum_{i=1}^n w_i p_i(x)} \, dx - 1 \tag{185}$$

$$\le \int \left( \sum_{i=1}^n w_i \right) \left( \sum_{i=1}^n w_i \frac{q(x)^2}{p_i(x)} \right) dx - 1 \tag{186}$$

$$= \sum_{i=1}^n w_i \left( \int \frac{q(x)^2}{p_i(x)} \, dx - 1 \right) = \sum_{i=1}^n w_i \chi^2(p_i \| q). \tag{187}$$

$\qquad\qquad\qquad\qquad\qquad\qquad\qquad\qquad\qquad\qquad\qquad\qquad\qquad\qquad\qquad\qquad\quad$ $\square$

**Lemma L.5.** *Suppose $p, \widetilde{p}$ are probability distributions such that $\frac{p}{\widetilde{p}} \le K$. Then for any probability distribution $q$,*

$$\chi^2(p \| q) \le K \chi^2(\widetilde{p} \| q) + K - 1. \tag{188}$$

*Proof.*

$$\chi^2(p \| q) = \int \frac{q^2}{p} - 1 \le K \int \frac{q^2}{\widetilde{p}} - 1 = K(\chi^2(\widetilde{p} \| q) + 1) - 1. \tag{189}$$

$\qquad\qquad\qquad\qquad\qquad\qquad\qquad\qquad\qquad\qquad\qquad\qquad\qquad\qquad\qquad\qquad\quad$ $\square$

## L.2 Chi-squared divergence calculations for log-concave distributions

We calculate the chi-squared divergence between log-concave distributions at different temperatures, and at different locations. In the gaussian case there is a closed formula (Lemma L.6). The general case is more involved (Lemmas L.11 and L.12), and the bound is in terms of the strong convexity and smoothness constants.

**Lemma L.6.** *For a matrix $\Sigma$, let $|\Sigma|$ denote its determinant. The $\chi^2$ distance between $N(\mu_1, \Sigma_1)$ and $N(\mu_2, \Sigma_2)$ is*

$$\chi^2(N(\mu_1, \Sigma_1) \| N(\mu_2, \Sigma_2)) \tag{190}$$

$$= \frac{|\Sigma_1|^{\frac{1}{2}}}{|\Sigma_2|} \left| (2\Sigma_2^{-1} - \Sigma_1^{-1}) \right|^{-\frac{1}{2}} \tag{191}$$

$$\cdot \exp\left( \frac{1}{2} (2\Sigma_2^{-1}\mu_2 - \Sigma_1^{-1}\mu_1)^T (2\Sigma_2^{-1} - \Sigma_1^{-1})^{-1}(2\Sigma_2^{-1}\mu_2 - \Sigma_1^{-1}\mu_1) + \frac{1}{2}\mu_1^T\Sigma_1^{-1}\mu_1 - \mu_2^T\Sigma_2^{-1}\mu_2 \right) - 1 \tag{192}$$

*In particular, in the cases of equal mean or equal variance,*

$$\chi^2(N(\mu, \Sigma_1), N(\mu, \Sigma_2)) = \frac{|\Sigma_1|^{\frac{1}{2}}}{|\Sigma_2|} \left| (2\Sigma_2^{-1} - \Sigma_1^{-1}) \right|^{-\frac{1}{2}} - 1 \tag{193}$$

$$\chi^2(N(\mu_1, \Sigma), N(\mu_2, \Sigma)) = \exp[(\mu_2 - \mu_1)^T \Sigma^{-1}(\mu_2 - \mu_1)]. \tag{194}$$

*Proof.*

$$\chi^2(N(\mu, \Sigma_1), N(\mu, \Sigma_2)) + 1 \tag{195}$$

$$= \frac{1}{(2\pi)^{\frac{d}{2}}} \frac{|\Sigma_1|^{\frac{1}{2}}}{|\Sigma_2|} \int_{\mathbb{R}^d} \exp\left[ -\frac{1}{2} \left( 2(x - \mu_2)^T \Sigma_2^{-1}(x - \mu_2) - (x - \mu_1)^T \Sigma_1^{-1}(x - \mu_1) \right) \right] dx \tag{196}$$

$$= \frac{1}{(2\pi)^{\frac{d}{2}}} \frac{|\Sigma_1|^{\frac{1}{2}}}{|\Sigma_2|} \int_{\mathbb{R}^d} \exp\left[ -\frac{1}{2} \left( x^T(2\Sigma_2^{-1} - \Sigma_1^{-1})x + 2x^T\Sigma_1^{-1}\mu_1 - 4x^T\Sigma_2^{-1}x - \mu_1^T\Sigma_1^{-1}\mu_1 + 2\mu_2^T\Sigma_2^{-1}\mu_2 \right) \right] dx \tag{197}$$

$$= \frac{1}{(2\pi)^{\frac{d}{2}}} \frac{|\Sigma_1|^{\frac{1}{2}}}{|\Sigma_2|} \int_{\mathbb{R}^d} \exp\left[ -\frac{1}{2}(x'^T(2\Sigma_2^{-1} - \Sigma_1^{-1})x' + c) \right] \tag{198}$$

$$x' := x - (2\Sigma_2^{-1} - \Sigma_1^{-1})^{-1}(\mu_1^T\Sigma_1^{-1} - 2\mu_2^T\Sigma_2^{-1}) \tag{199}$$

$$c := \frac{1}{2}(2\Sigma_2^{-1}\mu_2 - \Sigma_1^{-1}\mu_1)^T(2\Sigma_2^{-1} - \Sigma_1^{-1})^{-1}(2\Sigma_2^{-1}\mu_2 - \Sigma_1^{-1}\mu_1) + \frac{1}{2}\mu_1^T\Sigma_1^{-1}\mu_1 - \mu_2^T\Sigma_2^{-1}\mu_2 \tag{200}$$

Integrating gives the result. For the equal variance case,

$$c = \frac{1}{2}(2\mu_2 - \mu_1)\Sigma^{-1}(2\mu_2 - \mu_1) + \frac{1}{2}\mu_1\Sigma^{-1}\mu_1 - \mu_2\Sigma^{-1}\mu_2 = (\mu_2 - \mu_1)^T\Sigma^{-1}(\mu_2 - \mu_1)^T. \tag{201}$$

$\square$

The following theorem is essential in generalizing from gaussian to log-concave densities.

**Theorem L.7** (Hargé, [Har04]). *Suppose the $d$-dimensional gaussian $N(0, \Sigma)$ has density $\gamma$. Let $p = h \cdot \gamma$ be a probability density, where $h$ is log-concave. Let $g : \mathbb{R}^d \to \mathbb{R}$ be convex. Then*

$$\int_{\mathbb{R}^d} g(x - \mathbb{E}_p x)p(x)\,dx \le \int_{\mathbb{R}^d} g(x)\gamma(x)\,dx. \tag{202}$$

**Lemma L.8** ($\chi^2$-tail bound). *Let $\gamma$ be the distribution with probability density function $e^{-\frac{\kappa}{2}\|x\|^2}$. Then*

$$\forall y \ge \sqrt{\frac{d}{m}}, \quad \mathbb{P}_{x \sim \gamma}(\|x\| \ge y) \le e^{-\frac{\kappa}{2}\left(y - \sqrt{\frac{d}{\kappa}}\right)^2}. \tag{203}$$

•

*Proof.* By the $\chi_d^2$ tail bound in [LM00], for all $t \geq 0$,

$$\mathbb{P}_{x \sim \gamma}\left(\|x\|^2 \geq \frac{1}{\kappa}(\sqrt{d} + \sqrt{2t})^2\right) \leq \mathbb{P}_{x \sim \gamma}\left(\|x\|^2 \geq \frac{1}{\kappa}(d + 2(\sqrt{dt} + t))\right) \leq e^{-t} \quad (204)$$

$$\implies \forall y \geq \sqrt{\frac{d}{m}}, \quad \mathbb{P}_{x \sim \gamma}(\|x\| \geq y) \leq e^{-\left(\frac{\sqrt{\kappa}y - \sqrt{d}}{\sqrt{2}}\right)^2} = e^{-\frac{\kappa}{2}\left(y - \sqrt{\frac{d}{\kappa}}\right)^2} \quad (205)$$

$\square$

**Lemma L.9.** *Let $f : \mathbb{R}^d \to \mathbb{R}$ be a $\kappa$-strongly convex and $K$-smooth function and let $p(x) \propto e^{-f(x)}$ be a probability distribution. Let $x^* = \operatorname{argmin}_x f(x)$ and $\overline{x} = \mathbb{E}_p f(x)$. Then*

$$\|x^* - \overline{x}\| \leq \sqrt{\frac{d}{\kappa}}\left(\sqrt{\ln\left(\frac{K}{\kappa}\right)} + 5\right). \quad (206)$$

*Proof.* We establish both concentration around the mode $x^*$ and the mean $\overline{x}$. This will imply that the mode and the mean are close. Without loss of generality, assume $x^* = 0$ and $f(0) = 0$.

For the mode, note that by Lemma L.8, for all $r \geq \frac{d}{m}$,

$$\int_{\|x\| \geq r} e^{-f(x)}\, dx \leq \int_{\|x\| \geq r} e^{-\frac{1}{2}\kappa x^2} \leq \left(\frac{2\pi}{\kappa}\right)^{\frac{d}{2}} e^{-\frac{\kappa}{2}\left(r - \sqrt{\frac{d}{\kappa}}\right)^2} \quad (207)$$

$$\int_{\|x\| < r} e^{-f(x)}\, dx \geq \int_{\|x\| < r} e^{-\frac{1}{2}K x^2} \geq \left(\frac{2\pi}{K}\right)^{\frac{d}{2}}\left(1 - e^{-\frac{K}{2}\left(r - \sqrt{\frac{d}{\kappa}}\right)^2}\right). \quad (208)$$

Let $r = \sqrt{\frac{d}{\kappa}}\left(\sqrt{\ln\left(\frac{K}{\kappa}\right)} + 3\right)$. Then

$$\int_{\|x\| \geq r} e^{-f(x)}\, dx \leq \left(\frac{2\pi}{\kappa}\right)^{\frac{d}{2}} e^{-\frac{d}{2}\left(\ln\left(\frac{K}{\kappa}\right) + 2\right)} \leq \left(\frac{2\pi}{K}\right)^{\frac{d}{2}} e^{-d} \quad (209)$$

$$\int_{\|x\| < r} e^{-f(x)}\, dx \geq \left(\frac{2\pi}{K}\right)^{\frac{d}{2}}\left(1 - e^{-\frac{K}{2}\left(r - \sqrt{\frac{d}{\kappa}}\right)^2}\right) \quad (210)$$

$$\geq \left(\frac{2\pi}{K}\right)^{\frac{d}{2}}\left(1 - e^{-\frac{Kd}{2\kappa}\left(2 + \ln\left(\frac{K}{\kappa\kappa}\right)\right)}\right) \geq \left(\frac{2\pi}{K}\right)^{\frac{d}{2}}(1 - e^{-d}) \quad (211)$$

Thus

$$\mathbb{P}_{x \sim p}(\|x\| \geq r) = \frac{\int_{\|x\| \geq r} e^{-f(x)}\, dx}{\int_{\|x\| \geq r} e^{-f(x)}\, dx + \int_{\|x\| < r} e^{-f(x)}\, dx} \leq e^{-d} \leq \frac{1}{2}. \quad (212)$$

Now we show concentration around the mean. By adding a constant to $f$, we may assume that $p(x) = e^{-f(x)}$. Note that because $f$ is $\kappa$-smooth, $p$ is the product of $\gamma(x)$ with a log-concave function, where $\gamma(x)$ is the density of $N(0, \frac{1}{\kappa}I_d)$. note that by Hargé's Theorem L.7,

$$\int_{\mathbb{R}^d} \|x - \overline{x}\|^2\, p(x)\, dx \leq \int_{\mathbb{R}^d} \|x\|^2\, \gamma(x)\, dx = \frac{d}{\kappa}. \quad (213)$$

By Markov's inequality,

$$\mathbb{P}_{x \sim p}\left(\|x - \overline{x}\| \geq \sqrt{\frac{2d}{\kappa}}\right) = \mathbb{P}\left(\|x - \overline{x}\|^2 \geq \frac{2d}{\kappa}\right) \leq \frac{1}{2}. \quad (214)$$

Let $B_r(x)$ denote the ball of radius $r$ around $x$. By (212) and (214), $B_{\sqrt{\frac{d}{\kappa}}\left(\sqrt{\ln\left(\frac{K}{\kappa}\right)} + 3\right)}(x^*)$ and

$B_{\sqrt{\frac{2d}{\kappa}}}(\overline{x})$ intersect. Thus $\|\overline{x} - x^*\| \leq \sqrt{\frac{d}{\kappa}}\left(\sqrt{\ln\left(\frac{K}{\kappa}\right)} + 5\right)$. $\square$

**Lemma L.10** (Concentration around mode for log-concave distributions). *Suppose $f : \mathbb{R}^d \to \mathbb{R}$ is $\kappa$-strongly convex and $K$-smooth. Let $p$ be the probability distribution such that $p(x) \propto e^{-f(x)}$. Let $x^* = \operatorname{argmin}_x f(x)$. Then*

$$\mathbb{P}_{x \sim p}\left( \|x - x^*\|^2 \geq \frac{1}{\kappa}\left( \sqrt{d} + \sqrt{2t + d\ln\left(\frac{K}{\kappa}\right)} \right)^2 \right) \leq e^{-t}. \tag{215}$$

*Proof.* By (207) and (208),

$$\mathbb{P}_{x \sim p}(\|x\| \geq r) \leq \left(\frac{K}{\kappa}\right)^{\frac{d}{2}} e^{-\frac{\kappa}{2}\left(r - \sqrt{\frac{d}{\kappa}}\right)^2}. \tag{216}$$

Substituting in $r = \frac{1}{\sqrt{\kappa}}\left( \sqrt{d} + \sqrt{2t + d\ln\left(\frac{K}{\kappa}\right)} \right)$ gives the lemma. $\qquad\square$

**Lemma L.11** ($\chi^2$-divergence between translates). *Let $f : \mathbb{R}^d \to \mathbb{R}$ be a $\kappa$-strongly convex and $K$-smooth function and let $p(x) \propto e^{-f(x)}$ be a probability distribution. Let $\|\mu\| = D$. Then*

$$\chi^2(p(x - \mu)\|p(x)) \leq e^{\frac{1}{2}\kappa D^2 + KD\sqrt{\frac{d}{\kappa}}\left(\sqrt{\ln\left(\frac{K}{\kappa}\right)} + 5\right)}\left( e^{KD\sqrt{\frac{d}{\kappa}}} + KD\sqrt{\frac{4\pi}{\kappa}} e^{\frac{2KD\sqrt{d}}{\sqrt{\kappa}} + \frac{K^2 D^2}{2\kappa}} \right) - 1. \tag{217}$$

*Proof.* Without loss of generality, suppose $f$ attains minimum at 0, or equivalently, $\nabla f(0) = 0$. We bound

$$\chi^2(p(x - \mu)\|p(x)) + 1 = \int_{\mathbb{R}^d} \frac{e^{-2f(x)}}{e^{-f(x-\mu)}}\, dx = \int_{\mathbb{R}^d} e^{-f(x)} e^{f(x-\mu) - f(x)}\, dx \tag{218}$$

$$\leq \int_{\mathbb{R}^d} e^{-f(x)} e^{KD\|x\| + \frac{1}{2}\kappa D^2}\, dx \tag{219}$$

Note that because $f$ is $\kappa$-strongly convex, $p$ is the product of $\gamma(x)$ with a log-concave function, where $\gamma(x)$ is the density of $N(0, \frac{1}{\kappa}I_d)$. Let $\overline{x} = \mathbb{E}_{x \sim p}\, x$ be the average value of $x$ under $p$. Apply Hargé's Theorem L.7 on $g(x) = e^{KD\|x + \overline{x}\|}$, $p(x) = e^{-f(x)}$ to get that

$$\int_{\mathbb{R}^d} e^{-f(x)} e^{KD\|x\|}\, dx \leq \int_{\mathbb{R}^d} \gamma(x) e^{KD\|x + \overline{x}\|}\, dx \tag{220}$$

$$= e^{KD\|\overline{x}\|}\left( e^{KD\sqrt{\frac{d}{\kappa}}} + \int_{\sqrt{\frac{d}{\kappa}}}^{\infty} \mathbb{P}_{x \sim \gamma}(\|x\| \geq y) KD e^{KDy}\, dy \right) \tag{221}$$

where we used the identity $\int_{\mathbb{R}} f(x)p(x)\, dx = f(y_0) + \int_{y_0}^{\infty} \mathbb{P}_{x \sim p}(x \geq y) f'(y)\, dy$ when $f(x)$ is an increasing function. By Lemma L.8,

$$\forall y \geq \sqrt{\frac{d}{m}}, \quad \mathbb{P}_{x \sim \gamma}\left(\|x\| \geq y\right) \leq e^{-\frac{\kappa}{2}\left(y - \sqrt{\frac{d}{\kappa}}\right)^2} \tag{222}$$

$$\implies \int_{\sqrt{\frac{d}{\kappa}}}^{\infty} \mathbb{P}_{x \sim \gamma}(\|x\| \geq y) KD e^{KDy}\, dy \leq KD \int_{\sqrt{\frac{d}{\kappa}}}^{\infty} e^{-\frac{\kappa}{2}\left(y - \frac{d}{\kappa}\right)^2 + KDy}\, dy \tag{223}$$

$$= KD \int_{\sqrt{\frac{d}{\kappa}}}^{\infty} e^{-\frac{\kappa}{2}\left[\left(y - \frac{d}{\kappa}\right)^2 - \frac{2KD\sqrt{d}}{\kappa^{\frac{3}{2}}} - \frac{K^2 D^2}{\kappa^2}\right]}\, dy \tag{224}$$

$$= KD \int_{\sqrt{\frac{d}{\kappa}}}^{\infty} e^{-\frac{\kappa}{2}\left(y - \frac{d}{\kappa}\right)^2 + \frac{2KD\sqrt{d}}{\sqrt{\kappa}} + \frac{K^2 D^2}{2\kappa}}\, dy \tag{225}$$

$$\leq KD\sqrt{\frac{4\pi}{\kappa}} e^{\frac{2KD\sqrt{d}}{\sqrt{\kappa}} + \frac{K^2 D^2}{2\kappa}}. \tag{226}$$

Putting together (219), (221), and (226), and using Lemma L.9,

$$\chi^2(p(x-\mu)||p(x)) \leq e^{\frac{1}{2}\kappa D^2 + KD\sqrt{\frac{d}{\kappa}}\left(\sqrt{\ln\left(\frac{K}{\kappa}\right)}+5\right)}\left(e^{KD\sqrt{\frac{d}{\kappa}}} + KD\sqrt{\frac{4\pi}{\kappa}}e^{\frac{2KD\sqrt{d}}{\sqrt{\kappa}}+\frac{K^2D^2}{2\kappa}}\right).$$

(227)

$\square$

**Lemma L.12** ($\chi^2$-divergence between temperatures). *Let $f : \mathbb{R}^d \to \mathbb{R}$ be a $\kappa$-strongly convex and $K$-smooth function and let $p(x) \propto e^{-f(x)}$, $p_\beta(x) \propto e^{-\beta f(x)}$ be probability distributions. Suppose $\beta_1, \beta_2 > 0$ and $|\beta_2 - \beta_1| < \frac{\kappa}{K}$. Then*

$$\chi^2(p_{\beta_1}||p_{\beta_2}) \leq e^{\frac{1}{2}\left|1-\frac{\beta_1}{\beta_2}\right|\frac{Kd}{\kappa-K\left|1-\frac{\beta_1}{\beta_2}\right|}\left(\sqrt{\ln\left(\frac{K}{\kappa}\right)}+5\right)^2}\left(\left(1-\frac{K}{\kappa}\left|1-\frac{\beta_1}{\beta_2}\right|\right)\left(1+\left|1-\frac{\beta_{i-1}}{\beta_i}\right|\right)\right)^{-\frac{d}{2}} - 1.$$

(228)

*Proof.* Without loss of generality, suppose $f$ attains minimum at 0 (or equivalently, $\nabla f(0) = 0$), and $f(0) = 0$. We bound

$$\chi^2(p_{\beta_1}||p_{\beta_2}) + 1 = \frac{\int_{\mathbb{R}^d} e^{-\beta_1 f(x)}\, dx \int_{\mathbb{R}^d} e^{(\beta_1 - 2\beta_2)f(x)}\, dx}{\left(\int_{\mathbb{R}^d} e^{-\beta_2 f(x)}\, dx\right)^2}.$$

(229)

Let $\overline{x} = \mathbb{E}_{x \sim p_{\beta_2}} x$ be the average value of $x$ under $p_{\beta_2}$. Note that because $f$ is $m$-strongly convex, $e^{-\beta_1 f(x)}$ is the product of $\gamma(x)$ with a log-concave function, where $\gamma(x)$ is the density of $N\left(0, \frac{1}{\beta_2\kappa}I_d\right)$. Applying Hargé's Theorem L.7 on $g_1(x) = e^{(\beta_2-\beta_1)f(x+\overline{x})}$ and $g_2(x) = e^{(\beta_1-\beta_2)f(x+\overline{x})}$ to get

$$(229) \leq \int e^{(\beta_2-\beta_1)f(x+\overline{x})}\, d\gamma(x) \int e^{(\beta_1-\beta_2)f(x+\overline{x})}\, d\gamma(x).$$

(230)

Because $f$ is $m$-strongly convex and $M$-smooth, and $f(0) = 0$ is the minimum of $f$,

$$(230) \leq \frac{\int_{\mathbb{R}^d} e^{-|\beta_2-\beta_1|\frac{K}{2}\|x+\overline{x}\|^2}e^{-\frac{\beta_2\kappa}{2}\|x\|^2}\, dx \int_{\mathbb{R}^d} e^{-|\beta_2-\beta_1|\frac{\kappa}{2}\|x+\overline{x}\|^2}e^{-\frac{\beta_2\kappa}{2}\|x\|^2}\, dx}{\left(\int_{\mathbb{R}^d} e^{-\frac{\beta_2 m}{2}\|x\|^2}\, dx\right)^2}.$$

(231)

Using the identity

$$a\|x+\overline{x}\|^2 + b\|x\|^2 = (a+b)\|x\|^2 + 2a\langle x, \overline{x}\rangle + \frac{a^2}{a+b}\|\overline{x}\|^2 + \frac{ab}{a+b}\|\overline{x}\|^2 \tag{232}$$

$$= (a+b)\left\|x + \frac{a}{a+b}\overline{x}\right\|^2 + \frac{ab}{a+b}\|\overline{x}\|^2, \tag{233}$$

we get using Lemma L.9

$$(231) = \frac{e^{\frac{K\kappa|\beta_2-\beta_1|\beta_2}{2(\kappa\beta_2 - K|\beta_2-\beta_1|)}\|\overline{x}\|^2}\int_{\mathbb{R}^d} e^{\left(\frac{K}{2}|\beta_2-\beta_1|-\frac{\kappa}{2}\beta_2\right)\|x+\cdots\|^2}\, dx \cdot e^{\frac{-\kappa|\beta_1-\beta_2|\beta_2}{2\kappa(\beta_2-|\beta_2-\beta_1|)}\|\overline{x}\|^2}\int_{\mathbb{R}^d} e^{\left(-\frac{\kappa}{2}\beta_2-|\beta_2-\beta_1|\frac{\kappa}{2}\right)\|x+\cdots\|^2}\, dx}{\left(\int_{\mathbb{R}^d} e^{-\frac{\beta_2\kappa}{2}\|x\|^2}\, dx\right)^2}$$

(234)

$$\leq e^{\frac{|\beta_2-\beta_1|}{2}\frac{K\kappa\beta_2}{\kappa\beta_2 - K|\beta_2-\beta_1|}\|\overline{x}\|^2}\left(\frac{2\pi}{\kappa\beta_2 - K|\beta_2-\beta_1|}\right)^{\frac{d}{2}}\left(\frac{2\pi}{\kappa(\beta_2+|\beta_2-\beta_1|)}\right)^{\frac{d}{2}}\left(\frac{2\pi}{\kappa\beta_2}\right)^{-d}$$

(235)

$$\leq e^{\frac{1}{2}\left|1-\frac{\beta_1}{\beta_2}\right|\frac{Kd}{\kappa-K\left|1-\frac{\beta_1}{\beta_2}\right|}\left(\sqrt{\ln\left(\frac{K}{\kappa}\right)}+5\right)^2}\left(\left(1-\frac{K}{\kappa}\left|1-\frac{\beta_1}{\beta_2}\right|\right)\left(1+\left|1-\frac{\beta_{i-1}}{\beta_i}\right|\right)\right)^{-\frac{d}{2}}. \tag{236}$$

$\square$

**Lemma L.13** ($\chi^2$ divergence between gaussian and log-concave distribution). *Suppose that $p(x) \propto e^{-f(x-\mu)}$ is a probability density function, where $f$ is $\kappa$-strongly convex, $K$-smooth, and attains minimum at 0. Let $D = \|\mu\|$. Then*

$$\chi^2\left(p \| N\left(0, \frac{1}{K}I_d\right)\right) \leq \left(\frac{K}{\kappa}\right)^{\frac{d}{2}} e^{KD^2}. \tag{237}$$

*Proof.* We calculate

$$p(x) = \frac{e^{-f_0(x-\mu)}}{\int_{\mathbb{R}^d} e^{-f_0(u-\mu)}\, du} \geq \frac{e^{-\frac{K}{2}\|x-\mu\|^2}}{\int_{\mathbb{R}^d} e^{-\frac{\kappa}{2}\|u-\mu\|^2}\, du} = \left(\frac{\kappa}{2\pi}\right)^{\frac{d}{2}} e^{-\frac{K}{2}\|x-\mu\|^2}. \tag{238}$$

Then

$$\chi^2\left(p \| N\left(0, \frac{1}{K}I_d\right)\right) = \int_{\mathbb{R}^d} \frac{\left(\frac{K}{2\pi}\right)^d e^{-K\|x\|^2}}{p(x)}\, dx - 1 \tag{239}$$

$$\leq \left(\frac{K}{2\pi}\right)^{\frac{d}{2}} \left(\frac{2\pi}{\kappa}\right)^{\frac{d}{2}} \int_{\mathbb{R}^d} e^{-K\left(\|x\|^2 - \frac{1}{2}\|x-\mu\|^2\right)}\, dx \tag{240}$$

$$= \left(\frac{K}{\kappa}\right)^{\frac{d}{2}} \frac{K^{\frac{d}{2}}}{2\pi} \int e^{-\frac{K}{2}\|x+\mu\|^2 + K\|\mu\|^2}\, dx \tag{241}$$

$$\leq \left(\frac{K}{\kappa}\right)^{\frac{d}{2}} e^{KD^2}. \tag{242}$$

$\square$

## L.3 A probability ratio calculation

**Lemma L.14.** *Suppose that $f(x) = -\ln\left[\sum_{i=1}^n w_i e^{-\frac{\|x-\mu_i\|^2}{2}}\right]$, $p(x) \propto e^{-f(x)}$, and for $\alpha \geq 0$ let $p_\alpha(x) \propto e^{-\alpha f(x)}$, $Z_\alpha = \int_{\mathbb{R}^d} e^{-\alpha f(x)}\, dx$. Suppose that $\|\mu_i\| \leq D$ for all $i$. If $\alpha < \beta$, then*

$$\left[\int_A \min\{p_\alpha(x), p_\beta(x)\}\, dx\right] / p_\beta(A) \geq \min_x \frac{p_\alpha(x)}{p_\beta(x)} \geq \frac{Z_\beta}{Z_\alpha} \tag{243}$$

$$\frac{Z_\beta}{Z_\alpha} \in \left[\frac{1}{2} e^{-2(\beta-\alpha)\left(D + \frac{1}{\sqrt{\alpha}}\left(\sqrt{d} + 2\sqrt{\ln\left(\frac{2}{w_{\min}}\right)}\right)\right)^2}, 1\right]. \tag{244}$$

*Choosing $\beta - \alpha = O\left(\frac{1}{D^2 + \frac{d}{\alpha} + \frac{1}{\alpha}\ln\left(\frac{1}{w_{\min}}\right)}\right)$, this quantity is $\Omega(1)$.*

This is a special case of the following more general lemma.

**Lemma L.15.** *Suppose that $f(x) = -\ln\left[\sum_{i=1}^n w_i e^{-f_i(x)}\right]$, where $f_i(x) = f_0(x - \mu_i)$, $f_0$ is $\kappa$-strongly convex and $K$-smooth, $p(x) \propto e^{-f(x)}$, and for $\alpha \geq 0$ let $p_\alpha(x) \propto e^{-\alpha f(x)}$, $Z_\alpha = \int_{\mathbb{R}^d} e^{-\alpha f(x)}\, dx$. Suppose that $\|\mu_i\| \leq D$ for all $i$.*

*Let $C = D + \frac{1}{\sqrt{\alpha \kappa}}\left(\sqrt{d} + \sqrt{d\ln\left(\frac{K}{\kappa}\right) + 2\ln\left(\frac{2}{w_{\min}}\right)}\right)$. If $\alpha < \beta$, then*

$$\left[\int_A \min\{p_\alpha(x), p_\beta(x)\}\, dx\right] / p_\beta(A) \geq \min_x \frac{p_\alpha(x)}{p_\beta(x)} \geq \frac{Z_\beta}{Z_\alpha} \tag{245}$$

$$\frac{Z_\beta}{Z_\alpha} \in \left[\frac{1}{2} e^{-\frac{1}{2}(\beta-\alpha)KC^2}, 1\right]. \tag{246}$$

*If $\beta - \alpha = O\left(\frac{1}{K\left(D^2 + \frac{d}{\alpha\kappa}\left(1 + \ln\left(\frac{K}{\kappa}\right)\right) + \frac{1}{\alpha\kappa}\ln\left(\frac{1}{w_{\min}}\right)\right)}\right)$, then this quantity is $\Omega(1)$.*

*Proof.* Let $\widetilde{p}_\alpha(x) \propto \sum_{i=1}^n w_i e^{-\alpha f_0(x-\mu_i)}$ and $p \propto e^{-\alpha f_0(x)}$ be probability distributions.

By Lemma E.3 and Lemma L.10, since $\alpha f$ is $\alpha\kappa$-strongly, convex,

$$\mathbb{P}_{x \sim p}(\|x\| \geq C) \leq \frac{1}{w_{\min}} \mathbb{P}_{x \sim \widetilde{p}_\alpha}(\|x\| \geq C) \tag{247}$$

$$\leq \frac{1}{w_{\min}} \sum_{i=1}^n w_i \mathbb{P}_{x \sim \widetilde{p}_\alpha}(\|x\| \geq C) \tag{248}$$

$$\leq \frac{1}{w_{\min}} \sum_{i=1}^n w_i \mathbb{P}_{x \sim p}(\|x\|^2 \geq (C-D)^2) \tag{249}$$

$$= \frac{1}{w_{\min}} \mathbb{P}_{x \sim p} \left[ \|x\|^2 \geq \frac{1}{\alpha\kappa} \left( \sqrt{d} + \sqrt{d \ln\left(\frac{K}{\kappa}\right) + 2\ln\left(\frac{2}{w_{\min}}\right)} \right)^2 \right] \tag{250}$$

$$\leq \frac{1}{w_{\min}} \frac{w_{\min}}{2} = \frac{1}{2}. \tag{251}$$

Thus, using $f(x) \geq 0$,

$$\left[ \int_A \min\{p_\alpha(x), p_\beta(x)\} \, dx \right] / p_\beta(A) \geq \int_A \min\left\{\frac{p_\alpha(x)}{p_\beta(x)}, 1\right\} p_\beta(x) \, dx \Big/ p_\beta(A) \tag{252}$$

$$\geq \int_A \min\left\{\frac{Z_\beta}{Z_\alpha} e^{(\beta-\alpha)f(x)}, 1\right\} p_\beta(x) \, dx \Big/ p_\beta(A) \tag{253}$$

$$\geq \frac{Z_\beta}{Z_\alpha} \tag{254}$$

$$= \frac{\int e^{-\beta f(x)} \, dx}{\int e^{-\alpha f(x)} \, dx} \tag{255}$$

$$= \int_{\mathbb{R}^d} e^{(-\beta+\alpha)f(x)} p_\alpha(x) \, dx \tag{256}$$

$$\geq \int_{\|x\| \leq C} e^{(-\beta+\alpha)f(x)} p_\alpha(x) \, dx \tag{257}$$

$$\geq \frac{1}{2} e^{-(\beta-\alpha)\max_{\|x\| \leq C}(f(x))} \tag{258}$$

$$\geq \frac{1}{2} e^{-\frac{1}{2}(\beta-\alpha)MC^2}. \tag{259}$$

$\square$

### L.4 Other facts

**Lemma L.16.** *Let $(N_T)_{T \geq 0}$ be a Poisson process with rate $\lambda$. Then there is a constant $C$ such that*

$$\mathbb{P}(N_T \geq n) \leq \left(\frac{Cn}{T\lambda}\right)^{-n}. \tag{260}$$

*Proof.* Assume $n > T\lambda$. We have by Stirling's formula

$$\mathbb{P}(N_T \geq n) = e^{-\lambda T} \sum_{m=n}^{\infty} \frac{(\lambda T)^m}{m!} \tag{261}$$

$$\leq e^{-\lambda T} \frac{1}{n!} \frac{1}{1 - \frac{\lambda T}{n}} (\lambda T)^n \tag{262}$$

$$= e^{-\lambda T} O\left(n^{-\frac{1}{2}} \left(\frac{e\lambda T}{n}\right)^n\right) \tag{263}$$

$$\leq \left(\frac{Cn}{\lambda T}\right)^n \tag{264}$$

for some $C$, since $e^{-\lambda T} \geq e^{-n}$. $\qquad\qquad\qquad\qquad\qquad\qquad\qquad\qquad\qquad\qquad\qquad$ $\square$

## Footnotes

[1]We will implicitly assume this condition whenever we discuss Poincaré inequalities. Formally, we say that $g \in \mathcal{D}(\mathscr{E}_M)$ is in the Dirichlet domain of $\mathscr{E}_M$.