[Reviews · NeurIPS 2018]

Reviewer 1



In recent years, a lot of effort has been dedicated to the study of the discretization of the Langevin diffusion, known as the Langevin Monte Carlo (LMC) to sample from a distribution p. Previous works on LMC have mainly focused on log-concave densities to establish precise upper bounds on the number of iterations sufficient for the algorithm to be close to p (in total variation or Wasserstein distance). This article proposes to extend these polynomial bounds to a mixture of strongly log concave densities. For that purpose, the authors combine the Langevin diffusion with a simulated tempering that enables to mix between the modes of the distribution. The article has two main contributions. First, it suggests a new algorithm (simulated tempering combined with the Langevin diffusion) to sample from a mixture of strongly log-concave densities. Second, it provides polynomial bounds for the running time of the algorithm. I found the idea of this paper interesting but I found it very hard to read and understand. The authors should really make an effort to improve the presentation of their methods and the proof of their theoretical results. I did not understand at first reading the algorithm ! The authors do not provide any numerical illustrations of their algorithm. I think a simple example would be welcome. Other comments 1. line 44, typos: ”a log-concave distribution”, ”its density function” 2. line 47, what is a posterior distribution of the means for a mixture of gaussians ? 3. line 51, ”As worst-case results are prohibited by hardness results” please provide references. 4. line 73, typo ”The corresponding distribution e −f (x) are strongly log-concave distributions”. Do you mean e −f 0 (x) ? 5. Theorem 1.2. Please define the running time. 6. line 151, ”higher temperature distribution”, is β ∝ 1/T where T is the temperature ? Please clarify ”Using terminology from statistical physics, β = 1/τ is the inverse temperature” line 173 before this first statement line 151. 7. line 158, ”with stationary distribution.” an expression missing ?, e −β i f ? 8. line 167, typo, ”is to preserve” 9. Proposition 2.1. The discretization of the Langevin dynamics yields a non-reversible Markov chain, of invariant distribution close but different from p the target distribution. 10. line 195, Algorithm 1. x ← x − η 0 β i ∇f (x) + ..., confusion with n ← n + 1 and subsequent indices. Algorithm 2, what is δ ? 11. line 200, typo simulated.

Reviewer 2



Summary: This paper considers the problem of sampling from multi-modal distributions. Specifically they are interested in sampling from distributions that are (close to) mixtures of log-concave distributions with the same shape. The algorithm that they consider for this task is a simulated tempering algorithm which utilizes Langevin dynamics Markov chains at varying temperatures. They show that this algorithm can produce samples close to the desired distribution in polynomial-time. Quality: The paper appears to be technically sound, although I have not verified the proofs rigorously. Clarity: This submission is easy enough to read, although it is light on details. Since one of the major tools/contributions of the paper is the decomposition theorem in Appendix D, I would have preferred if some statement of this theorem had appeared in the submission itself. Originality: This paper appears to present the first provably efficient algorithm for sampling from mixtures of log-concave distributions using only gradient information of the log-pdf. Moreover, although decomposition theorems have previously been used to analyze Markov chains, the decomposition theorems presented in this paper appear to be novel as well. Significance: I think that this is a very interesting theoretical contribution to the study of sampling algorithms. Sampling from multi-modal distributions is an important problem in machine learning, and this paper makes progress in showing that a classical heuristic (simulated tempering) can be applied in this particular setting. Moreover, I think the techniques developed in this paper may be useful to others in this field. That said, I do not know of any practical situations in which there is a distribution that we want to sample from that (a) is a mixture of log-concave distributions with the same shape and (b) we have only access to the gradient of its log-pdf. I think this submission would be made significantly stronger with a discussion of such situations. Detailed comments: - In the last sentence of the abstract, it states “…this is the first result that proves fast mixing for multimodal distributions.” However, since there are many Markov chains that provably sample from multimodal distributions, I think you probably meant something to the effect of “…with only access to the gradient of the log-pdf.” - In lines 58-59 and 102-104, you discuss the lower bounds presented in Appendix K. I think it is important to clarify that you are talking about algorithms that have only access to the log-pdf and its gradient.

Reviewer 3



The paper studies the problem of sampling from multi-modal distributions using simulated tempering techniques combined with Langevin Monte Carlo. Though the assumption about mixture of strongly log-concave component with the same shape is very artificial, sampling from it with access only to its gradient oracle is still a non-trivial problem. More importantly, since the algorithm itself is not too specific for this model, I think it opens up a promising direction for sampling from more general multi-modal distributions. As far as a know, this is the first attempt towards mixing rate for multi-modal distributions that has polynomial dependence on everything. The techniques are very interesting: it is known that Langevin diffusion converges fast to stationary at very high temperature, even with multi-modal distributions. This paper makes use of this fact by combining it a meta Markov Chain which moves between partitions at the highest temperature, with fast mixing Langevin algorithm for partitions. To make simulated tempering possible, partition functions are estimated in an iterative way for different temperature levels. The mixing rate estimates for both chains are based on spectral gap estimates. Besides, one thing I'm not sure is that, even if you're given only gradient oracles for the log density, is it possible to estimate those components so that such heavy mechanism is not needed? For example, if we uniformly sample a lot of starting points and run gradient descent starting from them to find the local minima of -log(p), the modes of each mixture components can be recovered. Then the sampling problem becomes almost trivial for the Gaussian case, and simple tricks seem to work in strongly-log-concave component case. If this simple approach is not excluded, the contribution of this paper might be eclipsed, which explains my score. The discretization of Langevin diffusion without Metropolis adjustment for mixing rate proof for each component, while the simulated tempering chain is run with that step. Are there particular reason for this?